# Radiation Without Borders: Unraveling Bystander and Non-Targeted Effects in Oncology

**DOI:** 10.3390/cells14221761

**Published:** 2025-11-11

**Authors:** Madhi Oli Ramamurthy, Poorvi Subramanian, Sivaroopan Aravindan, Loganayaki Periyasamy, Natarajan Aravindan

**Affiliations:** Division of Molecular Oncology, Department of Physiological Sciences, Oklahoma State University, Stillwater, OK 74078, USA; madhioli.ramamurthy@okstate.edu (M.O.R.); poorvi.subramanian@okstate.edu (P.S.); roopanaravindan@gmail.com (S.A.); loganayaki.periyasamy@okstate.edu (L.P.)

**Keywords:** radiotherapy, radiation bystander effect, abscopal effects of radiation, non-targeted radiation response, effects of tumor targeted radiotherapy

## Abstract

**Highlights:**

**What are the main findings?**

**What are the implications of the main findings?**

**Abstract:**

Radiotherapy (RT) remains a cornerstone of cancer treatment, offering spatially precise cytotoxicity against malignant cells. However, emerging evidence reveals that ionizing radiation (IR) exerts biological effects beyond the targeted tumor volume, manifesting as radiation bystander effects (BEs) and other non-targeted effects (NTEs). These phenomena challenge the traditional paradigm of RT as a localized intervention, highlighting systemic and long-term consequences in non-irradiated tissues. This comprehensive review synthesizes molecular, cellular, and clinical insights about BEs, elucidating the complex intercellular signaling networks gap junctions, cytokines, extracellular vesicles, and oxidative stress that propagate damage, genomic instability, and inflammation. We explore the role of mitochondrial dysfunction, epigenetic reprogramming, immune modulation, and stem cell niche disruption in shaping BEs outcomes. Clinically, BEs contribute to neurocognitive decline, cardiovascular disease, pulmonary fibrosis, gastrointestinal toxicity, and secondary malignancies, particularly in pediatric and long-term cancer survivors. The review also evaluates countermeasures including antioxidants, COX-2 inhibitors, exosome blockers, and FLASH RT, alongside emerging strategies targeting cfCh, inflammasomes, and senescence-associated secretory phenotypes. We discuss the dual nature of BEs: their potential to both harm and heal, underscoring adaptive responses and immune priming in specific contexts. By integrating mechanistic depth with translational relevance, this work posits that radiation BEs are a modifiable axis of RT biology. Recognizing and mitigating BEs is imperative for optimizing therapeutic efficacy, minimizing collateral damage, and enhancing survivorship outcomes. This review advocates for a paradigm shift in RT planning and post-treatment care, emphasizing precision, personalization, and systemic awareness in modern oncology.

## 1. Radiation in Cancer Treatment

Cancer is one of the most formidable public health challenges in the United States, with the American Cancer Society projecting nearly two million new diagnoses in 2025 [1]. This predictable burden not only reflects the persistent incidence of malignancy across the population but also underscores the critical need for efficacious and adaptable therapeutic strategies capable of addressing diverse tumor types and clinical presentations [2]. Within this therapeutic continuum, radiation therapy (RT) has long been established as a cornerstone of cancer management, serving as a fundamental component in both curative and palliative regimens [3]. More than half of all individuals diagnosed with cancer are anticipated to receive RT at some point during their clinical course, attesting to its broad applicability and clinical relevance across early-stage, locally advanced, and metastatic disease settings [1]. The primary mechanism by which RT exerts its antineoplastic effect involves the induction of irreparable DNA damage in malignant cells, leading to proliferative arrest, apoptosis, or mitotic catastrophe [4]. A defining feature of RT is its capacity to deliver cytotoxic energy with spatial selectivity, thereby confining damage predominantly to the tumor volume while sparing surrounding normal tissues [5]. This spatial precision has been significantly improved by continuous advancements in imaging modalities, treatment planning systems, and radiation delivery technologies [6]. The integration of these innovations has resulted in measurable clinical gains, including improved locoregional tumor control, enhanced overall survival (OS) in select contexts, and a reduction in treatment-related toxicities [7,8]. The consequence is a more favorable therapeutic ratio that enables clinicians to achieve oncologic control while minimizing long-term complications and enhancing post-treatment quality of life [9]. The clinical application of RT spans a wide spectrum of oncologic indications. It may serve as a definitive monotherapy in early-stage malignancies or be incorporated into multimodal strategies alongside surgical resection and systemic therapies [1,10].

In the neoadjuvant setting, RT is often employed to reduce tumor volume and vascularity, thereby improving surgical resectability and local control. In the adjuvant context, it facilitates the eradication of residual microscopic disease, reducing the risk of locoregional recurrence and contributing to improved disease-free survival (DFS) [11]. Increasingly, the biological effects of RT are being recognized as extending beyond direct tumor cytotoxicity [12]. Accumulating evidence demonstrates that RT can modulate the tumor microenvironment (TME) by altering immune cell infiltration, promoting vascular normalization, and inducing pro-inflammatory and stress-related signaling cascades [13]. These non-canonical effects have reinvigorated interest in RT as a modulator of anti-tumor immunity and as a synergistic partner in combination regimens involving immunotherapy and molecularly targeted agents [14]. In the palliative setting, RT continues to provide substantial clinical benefit by alleviating tumor-related symptoms such as pain, hemorrhage, and obstruction in patients with advanced or metastatic disease [15]. Its judicious application can significantly enhance patient comfort and quality of life, particularly when curative intervention is no longer feasible [16]. The selection of the appropriate RT modality in each clinical scenario is influenced by multiple factors, including tumor location, histological subtype, proximity to radiosensitive structures, and the patient’s overall performance status [17].

Technological evolution in RT has given rise to a broad and sophisticated armamentarium of modalities [18]. These approaches may be broadly categorized into conformal external beam techniques, radionuclide-based internal therapies, and precision-guided radiation delivery systems [19]. Conformal external beam techniques, such as three-dimensional conformal RT (3D-CRT), utilize volumetric imaging and computer-assisted planning to generate radiation fields that precisely match the tumor’s anatomical contours [20]. This has been particularly effective in malignancies located in anatomically complex regions such as the central nervous system (CNS), head and neck, liver, and prostate [19,21,22]. Intensity-modulated RT (IMRT) and image-guided RT (IGRT) represent further refinements of this approach, incorporating dynamic beam modulation and real-time imaging, respectively, to enhance precision and spare adjacent normal tissues [5,19]. Volumetric modulated arc therapy (VMAT) advances these principles by enabling modulated dose delivery during continuous gantry rotation, allowing for highly conformal distributions in geometrically challenging tumor sites [19,23]. Internal radiation approaches, often classified as radionuclide-based therapies, offer an alternative mechanism by which IR is delivered in close proximity to tumor tissues [19,24,25]. Brachytherapy, the prototypical example, involves the placement of sealed radioactive sources directly within or adjacent to the tumor and is widely used in the treatment of gynecologic, prostate, and breast cancers [19,26,27]. Intraoperative RT (IORT) allows for the delivery of a high dose of radiation during surgery, typically targeting the tumor bed while minimizing exposure to uninvolved tissues, a strategy that is especially beneficial in gastrointestinal malignancies [19,28,29]. Stereotactic radiosurgery (SRS), a highly focused, high-dose modality, delivers ablative radiation to intracranial lesions such as gliomas and skull base tumors in one or a few sessions [19,30,31].

Concurrently, external beam RT technologies have undergone transformative innovation. Proton beam therapy (PBT), for instance, exploits the Bragg peak phenomenon to deposit the majority of the radiation dose at a specific tissue depth, thereby sparing healthy tissue distal to the tumor [19,32,33]. This unique dosimetric property makes PBT particularly advantageous in pediatric oncology and in cases where tumors are located adjacent to critical structures [19,34]. The integration of magnetic resonance imaging with linear accelerators has led to the development of MRI-guided RT platforms, which enable real-time adaptive treatment planning based on anatomical changes during therapy, offering unprecedented precision in soft tissue targeting [19,35]. Stereotactic body RT (SBRT), another critical modality, delivers ablative doses over a limited number of fractions and is increasingly favored for treating tumors in surgically inaccessible or anatomically constrained sites such as the lung, liver, pancreas, and adrenal glands [19,36]. Complementing these advances is the emergence of ultra-high dose rate techniques, most notably FLASH RT [37]. This modality involves the delivery of radiation doses within milliseconds at dose rates several orders of magnitude higher than conventional RT [36]. Preclinical studies suggest that FLASH RT may differentially spare normal tissues while preserving, or potentially enhancing, tumor control [38]. This represents a paradigm shift in RT, opening new avenues for minimizing toxicity while maintaining therapeutic potency.

Despite these advances, the clinical use of RT is accompanied by definable challenges. Accumulating data have illuminated the risk of unintended consequences following exposure to IR, including off-target effects on non-irradiated tissues (Table 1) [39]. Surviving cancer cells may undergo epigenetic, transcriptional and molecular reprogramming that confers resistance to subsequent therapies [40]. Of particular concern is the long-recognized risk of radiation-induced secondary malignancies, a complication of particular relevance in younger patients and long-term survivors treated with high cumulative doses [41]. These risks highlight the need for continued investigation into radiobiological mechanisms, dose optimization strategies, and the development of predictive biomarkers for response and toxicity [42].

Among these mechanisms, non-targeted effects (NTEs) represent a paradigm shift in our understanding of radiobiology, extending its impact beyond directly irradiated cells to neighboring and even distant tissues [39]. While mechanistic insights into RT have advanced substantially through cell culture-based investigations, it is crucial to recognize the limitations of these models in predicting real-world outcomes [43,44]. Foundational research on radiation-induced NTEs has predominantly relied on in vitro approaches, which offer controlled environments for dissecting molecular and cellular mechanisms [43,44]. However, a growing body of evidence demonstrates that results observed in vitro often diverge from those in vivo or in clinical settings due to the complex interplay of stromal elements, immune response, and heterogeneous tissue architecture present in living organisms [43,44]. These discrepancies underscore the need for careful interpretation of preclinical data and highlight ongoing challenges in translating bench side discoveries into clinically relevant strategies [43,44]. Accordingly, understanding the relevance and limitations of each experimental model system is essential for accurately contextualizing the impact of radiation NTEs in cancer therapy and for developing predictive frameworks that inform patient outcomes and treatment safety [43,44]. Considering these critical insights, this review explores the indirect and off-target consequences of radiation exposure, particularly RT-induced BEs and other NTEs, to dissect their molecular underpinnings and relevance to tumor response and normal tissue injury. By consolidating key evidence across diverse experimental models and cancer types, it underscores the biological and clinical importance of BEs and NTEs, positioning them as critical determinants of RT outcomes and long-term patient effects.

**Table 1 cells-14-01761-t001:** Class of risks associated with off-target effects of radiation.

Risk Category	Examples	Typical Onset	Ref.
Acute toxicities	Dermatitis, mucositis, diarrhea, nausea, fatigue, skin desquamation, oral ulcers, hair loss, low blood counts	During/soon after RT	[45,46,47]
Delayed toxicities	Pulmonary fibrosis, enteritis, xerostomia, chronic diarrhea, lymphedema, fibrosis, osteoradionecrosis, hypothyroidism, telangiectasia	Month-years post-RT	[46,48,49]
Neurocognitive effects	Memory loss, impaired processing, attention deficits, executive dysfunction	Late	[46,47,49]
Secondary cancers	Sarcomas, leukemias, carcinomas	Years/decades later	[47]
Normal tissue toxicities	Cardiotoxicity, pneumonitis, hypothyroidism, chronic cystitis, infertility, bowel/bladder dysfunction	Acute/Late	[46,47,48,49]
Secondary cellular responses	Genomic instability, chronic inflammation	Acute/Late	[49,50]
Radio resistance	Increased recurrence, molecular mayhem	Any time post-RT	[49,50]
Immunosuppression	Lymphopenia, increased infection risk, impaired T-cell function	Acute/Late	[47,49,50]
Musculoskeletal effects	Muscle fibrosis, joint stiffness, limited mobility, and bone fractures	Late	[46,47,49]
Lymphedema	Swelling of neck, arms, or legs; heavy/achy limbs	Early/Late	[47]
Skin changes	Redness, dryness, pigmentation changes, ulceration, chronic scarring	Acute/Late	[47]
Gastrointestinal effects	Nausea, diarrhea, proctitis, malabsorption, rectal bleeding, pain	Acute/Late	[45,46,47]
Genitourinary/reproductive effects	Bladder irritation, cystitis, incontinence, infertility, erectile dysfunction, menstrual changes	Acute/Late	[46,47,49]
Fatigue	Daily tiredness, weakness, reduced activity levels	Acute/Late	[47]
Psychosocial effects	Depression, anxiety, and body image disturbance	Any phase	[47,48,51]
Rare, serious complications	Tissue necrosis, catastrophic bleeding, and organ failure	Late	[46,48]

## 2. Unintended and Non-Targeted Biological Effects of RT: Radiation Bystander Effects (BEs)

Radiation BEs represent a critical inflection point in radiation medicine by showing that the biological consequences of IR extend beyond cells directly traversed by radiation tracks [52]. Instead, neighboring non-irradiated cells display diverse responses mediated by intercellular communication and soluble factors [53]. First reported by Nagasawa and Little in 1992, who observed sister chromatid exchanges in non-irradiated human fibroblasts co-cultured with irradiated cells [54], radiation BEs have since been validated across mammalian and non-mammalian models [55,56]. This phenomenon is now recognized as a key component of NTEs with important implications for both therapeutic efficacy and normal tissue toxicity (Figure 1) [57].

Functionally, radiation BEs are driven by multiple interconnected pathways [58]. Irradiated cells release a repertoire of signaling molecules, including pro-inflammatory cytokines (e.g., TNF-α, IL-6, TGF-β), reactive oxygen species (ROS), reactive nitrogen species (RNS), and nitric oxide (NO), which can diffuse to neighboring cells and induce oxidative stress and DNA damage [59]. Gap junction intercellular communication (GJIC), particularly via connexin43 channels, facilitates the direct transfer of secondary messengers such as calcium ions and small metabolites, further propagating stress signals to adjacent non-irradiated cells [60]. Extracellular vesicles (EVs), including exosomes and microvesicles, have also been implicated in transmitting bystander signals by delivering bioactive cargoes such as microRNAs, proteins, and damaged DNA fragments [61,62,63]. Together, these pathways activate downstream effectors such as p53, NFκB, and MAPK signaling cascades, culminating in endpoints including genomic instability, micronuclei formation, apoptosis, senescence, and altered transcriptional profiles [64]. The interplay between GJIC and soluble factor signaling creates a complex communication network that underpins the spatial and temporal dynamics of radiation BEs and shapes cellular stress responses and fate decisions. 

Key signaling pathways include NFκB, which drives pro-inflammatory cytokine and survival gene transcription [65], p53, which orchestrates DNA damage responses such as cell cycle arrest, apoptosis, and senescence; and MAPK cascades (ERK, JNK, p38), which regulate proliferation, apoptosis, and stress adaptation. Hypoxia-inducible factor 1-alpha (HIF-1α) stabilization under oxidative stress modulates genes related to metabolism and angiogenesis. Calcium signaling through gap junctions contributes to bystander signal propagation [66]. ROS and RNS produced in irradiated and bystander cells amplify oxidative stress, perpetuating DNA damage and signaling activation [67]. Furthermore, inflammasome activation and subsequent cytokine release link bystander signaling to innate immune responses [68]. Collectively, these pathways balance survival and death, shaping tissue responses beyond irradiated cells while driving epigenetic and transcriptional reprogramming that reinforces the bystander phenotype in non-irradiated populations.

Radiation BEs are tightly linked to epigenetic and transcriptional reprogramming in bystander cells [69]. Bystander signaling induces global DNA hypomethylation alongside gene-specific hypermethylation, altering transcriptional landscapes and promoting genomic instability [70]. Histone modifications, including acetylation and methylation at H3 and H4 residues, affect chromatin accessibility. MicroRNAs transported by EVs, such as miR-21 and miR-1246, regulate genes involved in apoptosis, proliferation, and immune responses. Persistent NFκB activation sustains transcription of pro-inflammatory and stress response genes, while p53 modulates DNA repair and cell fate, jointly maintaining the bystander phenotype [64]. Long-term transcriptional changes include upregulation of oxidative stress-related genes (e.g., SOD2, GPX1), DNA repair genes (e.g., XRCC1), and cytokine signaling components, which can persist through multiple cell generations [71]. Emerging data suggest these epigenetic alterations may be heritable via mitosis, implicating radiation BEs in radiation-induced genomic instability (RIGI) [72,73]. Non-coding RNAs and altered RNA editing further regulate bystander responses, ref. [74] illustrating the complex rewiring of gene networks in tumor and normal tissues that intersects with immune modulation.

The immune system plays a central role in amplifying and modulating radiation BEs, acting as both mediator and target of bystander signals. Irradiated cells release damage-associated molecular patterns (DAMPs), including HMGB1 and ATP, which activate pattern recognition receptors such as Toll-like receptors (TLRs) on immune cells, triggering inflammatory cascades [75]. This response induces pro-inflammatory cytokines (IL-6, IL-8, TNF-α) that recruit and/or activate immune cells and act as bystander mediators [52]. Macrophages and dendritic cells exposed to these signals undergo phenotypic changes that sustain inflammation and propagate bystander signaling [61]. Natural killer (NK) cells and cytotoxic T-lymphocytes may be activated by irradiated or bystander-altered antigen-presenting cells, linking radiation BEs to systemic immune modulation [76]. Under certain conditions, radiation BEs enhances antitumor immunity by increasing the visibility of tumor antigens and promoting immune infiltration [77]. Conversely, chronic bystander signaling may drive immunosuppressive pathways, including TGF-β release and regulatory T-cell recruitment, fostering radiation resistance and immune evasion [78]. Studies in immunocompromised mouse models demonstrate markedly reduced bystander responses, highlighting the necessity of intact immune signaling for full radiation BEs expression [79]. These findings position the immune system as a dynamic amplifier of bystander signaling, shaping local and systemic outcomes, and underscoring the importance of distinguishing among various non-targeted radiation effects.

Distinguishing radiation BEs from related phenomena such as the abscopal and cohort effects is essential. The abscopal effects refers to tumor regression at distant sites mediated primarily through systemic immune activation [39,43,80]. Localized irradiation combined with immune checkpoint inhibitors has demonstrated T cell-dependent abscopal responses [43,81]. While radiation BEs and the abscopal effects share non-irradiated tissue involvement, radiation BEs operates locally within tissue microenvironments, whereas the abscopal effect involves systemic immune cell trafficking [39,43,80]. The cohort effects, though less rigorously defined, describes population-level dynamics where irradiated cells influence neighboring unirradiated cells within the same tissue, overlapping conceptually with BEs but emphasizing emergent population responses [43,80]. Another subtype involves the malignant transformation of non-irradiated cells triggered by oncogenic factors or genomic material released from irradiated cells, a process observed in studies of radiation BEs and RIGI [82]. Additionally, NTEs exhibit hallmarks that distinguish them from conventional targeted-IR effects. Key hallmarks distinguish NTEs from direct effects of radiation, including delayed biological manifestations, abscopal effects, and the RIGI in the progeny of irradiated cells [83]. These characteristics underscore radiation BEs’s relevance to both oncology and normal tissue biology.

Extensive literature has established the role of secreted factors in radiation BEs, notably demonstrated by clonogenic death induced in naïve cells exposed to medium from irradiated cultures [44,84]. Subsequent studies elucidated oxidative and inflammatory signaling pathways and the TME’s role in modulating radiation BEs [85,86]. Importantly, radiation BEs contributes to both beneficial effects, such as anti-tumor immunity, and detrimental outcomes, including therapy resistance, secondary malignancies, and delayed normal tissue toxicity [87,88]. Appreciating radiation BEs as part of the wider NTEs spectrum is essential for advancing RT, balancing anti-tumor efficacy with protection of normal tissues, and mitigating long-term adverse effects.

## 3. Molecular Signaling Mechanisms That Coordinate Radiation BEs

The concept of radiation BEs underscores the capacity of irradiated cells to elicit complex biological responses in neighboring or distant non-irradiated cells. This phenomenon is driven by diverse yet intricate intercellular signaling networks comprising secreted cytokines, ROS, RNS, EVs, and direct gap junctional communication, etc. Key Radiation BEs triggered signaling flowthroughs that converge and coordinate the cause-effects viz., DNA damage, inflammation, carcinogenesis, resistance neurodegeneration etc., include:

### 3.1. Immediate Early, Stress and Survival Signaling

The propagation of radiation-induced damage beyond directly irradiated cells relies on an intricate network of intercellular signaling pathways [89]. At the frontline of this communication are gap junctions, composed largely of connexin 43 proteins, which facilitate the direct cytoplasmic exchange of small molecules such as calcium ions and ROS [90,91]. This direct molecular trafficking ensures rapid synchronization of stress signals across adjoining cells, effectively extending the irradiative insult beyond the initially targeted population [89]. Conversely, irradiated cells secrete a complex milieu of soluble factors and cytokines, including TNF-α, TGF-β, IL-6, and VEGF [91,92,93,94,95,96,97]. These soluble mediators diffuse in the tissue microenvironment, binding cognate receptors on neighboring cells and activating established signaling cascades such as NFκB, JAK-STAT, and MAPK [98]. To that end, we have shown that clinical RT triggered NFκB in cancer cells leads to the TNFα, IL1α, cMYC and SOD2 dependent paracrine signaling and TNFR1-dependent NFκB mediated NTEs in bystander cells [99]. Crucially we unveiled that radiation bystander response through NFκB is not a general effect, rather inflicts a function-specific signaling flow-through: TNFα-dependent survival advantage [99,100]; hTERT-dependent clonal expansion [101]; and MMP9-Erk-dependent tumor dissemination [102]. More importantly our preclinical in vivo studies affirmed that the tumor targeted RT exerted NFκB-mediated abscopal radiation response in the distal heart and brain tissues [103]. These pathways regulate transcriptional programs that modulate inflammation, apoptosis, and survival, thereby tailoring the downstream biological response according to cell type, presence or absence of other stress signaling molecules, intrinsic cellular state, organismal factor, and radiation dose [44,104].

Adding to the complexity, EVs and exosomes have emerged as critical vehicles for long-range signaling [105]. These nano-sized vesicles encapsulate bioactive cargoes like miRNAs (e.g., miR-21), proteins, and damaged DNA fragments, allowing the transfer of complex regulatory information even to distant, non-adjacent cells, [74,106]. The role of EVs is particularly exciting as it bridges classical paracrine signaling including TLR4, MAPK/ERK, AKT, etc., and systemic effects, providing insights into how localized radiation can evoke responses in remote tissues [105,107]. In addition to the exosomes, tunneling nanotubes (TNTs) serve as physical conduits for direct cytoplasmic exchange of organelles and macromolecules such as mitochondria and stress-signaling proteins [108,109]. TNTs extend the range of intercellular communication, such as the Cx43 associated pathways, ATP release and P2 purinergic signaling, autophagy, etc., facilitating stress propagation with distinct kinetics and selectivity compared to diffusion-based signaling [61,110]. The multifaceted nature of these signaling mechanisms is heightened by the involvement of purinergic signaling, where ATP and its metabolites are released from damaged or stressed cells, activating P2X/P2Y receptors to induce calcium waves and secondary messenger cascade [111,112]. Simultaneously, gaseous molecules known as gasotransmitters, including NO, carbon monoxide (CO), and hydrogen sulfide (H_2_S), rapidly diffuse and modulate redox states and vascular responses, contributing to the dynamic intercellular milieu during radiation-induced stress [12]. Together, these diverse signaling modes converge to form a highly coordinated intercellular network that operates in a spatiotemporally regulated manner, seamlessly integrating direct cell-to-cell communication with paracrine and systemic signals [86]. This complex orchestration sets the stage for downstream intracellular events, which further amplify and modulate the BEs.

### 3.2. Redox Imbalance and Metabolic Rewiring

A key consequence of this intercellular signaling network is the amplification of oxidative stress, which acts not only as a damaging agent but also as a crucial second messenger in radiation BEs [85,100]. The transfer of ROS and RNS across gap junctions and via EVs triggers redox-sensitive pathways in bystander cells, sustaining and amplifying stress signals [85,100]. Mitochondria are pivotal hubs in this process [113]. Radiation-induced mitochondrial dysfunction leads to enhanced production of ROS, mitochondrial DNA (mtDNA) release, and impaired electron transport chain activity [89,90,91,92,109,113,114,115]. The persistence of dysfunctional mitochondria, compounded by compromised mitophagy mechanisms (e.g., PINK1/PARKIN pathways), ensure sustained oxidative stress that reinforces paracrine signaling loops [116,117]. This oxidative milieu drives substantial metabolic reprogramming in bystander cells [93,94]. There is a shift toward glycolytic metabolism, evidenced by upregulated enzymes like GAPDH and LDHA, with concomitant lactate production that acidifies the microenvironment and fosters inflammatory signaling [95]. Energy-sensing pathways such as AMPK and mTOR respond dynamically to these metabolic perturbations, adjusting cellular survival, growth, and immune functions in a context-dependent manner [59,95,96,97,118]. Moreover, lipid metabolism reconfigures during radiation stress, with bioactive lipid mediators such as ceramide and sphingosine-1-phosphate (S1P) acting as potent signals for apoptosis, inflammation, and ferroptosis, a regulated form of iron-dependent lipid peroxidation [85,119,120]. Notably, immune cell subsets such as CD8^+^ T cells have been implicated in modulating lipid-peroxide transfers, illustrating the interface between metabolism and immune regulation within BEs [116]. Thus, oxidative stress and metabolic rewiring do not merely represent downstream effects but function as integral components that propagate, modulate, and contextualize radiation bystander signaling, linking the extracellular environment with intracellular fate decisions.

### 3.3. DNA Damage, Epigenetics, and Chromatin Remodeling

Following the initial intercellular communications and redox perturbations, bystander cells often exhibit hallmarks of genomic stress traditionally associated with direct irradiation. Remarkably, DNA damage response (DDR) pathways are robustly activated even in non-irradiated cells, emphasizing the potency of transmitted signals [99,117]. Key DDR kinases such as ATM and ATR detect damage markers, including phosphorylated H2AX (γH2AX), and initiate a cascade involving checkpoint proteins like p53 and p21 to orchestrate repair, cell cycle arrest, or apoptosis [68,74,90,120,121,122,123,124,125]. This activation stems not solely from incidental ROS-induced lesions but also from extrinsic signaling via cytokines, EV-delivered nucleic acids, and possibly cell-free chromatin particles internalized by bystander cells [117]. The reception of such DNA fragments may directly challenge genome integrity, triggering chromosomal aberrations and contributing to RIGI, a driver of carcinogenesis and long-term tissue dysfunction [126,127]. Layered onto these acute DDR signals are profound epigenetic modifications shaping sustained cellular phenotypes [99]. Radiation and bystander signaling modulate DNA methylation patterns, often hypomethylation globally with site-specific hypermethylation, alter histone post-translational modifications, and induce reorganization of chromatin architecture through chromatin remodeling complexes [92,121,128,129,130]. These modifications affect gene expression programs central to cell cycle control, apoptosis, and differentiation.

Progressively, microRNAs including miR-21, let-7 family etc., long non-coding RNAs (e.g., HOTAIR), and circular RNAs have emerged as key regulators of these epigenetic landscapes [104,111,131,132]. Encapsulated within EVs or expressed endogenously, they influence DNA repair efficiency, apoptotic thresholds, and inflammatory responses [132]. In addition, RNA editing mechanisms such as adenosine-to-inosine (A-to-I) editing modulate innate immune sensing and stress signaling, further influencing bystander behavior [111,133]. These genomic and epigenomic alterations collectively act as a molecular memory of irradiation events, propagating biological outcomes across cell generations and potentially contributing to transgenerational effects [111]. Furthermore, recent discoveries around the disruption of liquid–liquid phase separation (LLPS), membraneless organelles that compartmentalize nuclear and cytoplasmic biomolecules, suggest that radiation stress perturbs fundamental principles of nuclear organization, influencing transcription, RNA metabolism, and DNA repair fidelity. Together, DNA damage signaling intertwined with epigenetic remodeling forms a durable axis through which radiation BEs manifests as persistent changes in cell fate and tissue homeostasis.

### 3.4. Immune Response and Inflammation

Irradiated and bystander cells secrete a complex milieu of pro- and anti-inflammatory cytokines and chemokines, including IL-6, TNF-α, IL-1β, IL-10, and TGF-β [85,86,100,110,128,134,135,136,137,138]. These factors recruit and reprogram immune cells such as macrophages, microglia, T-cells, and dendritic cells, shaping the local immune landscape [89,125,134,138,139,140,141]. Macrophages, for instance, may polarize toward pro-inflammatory (M1) or reparative (M2) phenotypes, influencing tissue remodeling and repair [111,132]. A particularly intriguing aspect is the concept of trained immunity, where innate immune cells undergo durable epigenetic and metabolic reprogramming following radiation exposure [95,128]. This reprogramming heightens or dampens responses to subsequent stimuli, contributing to either aggravation or resolution of inflammation [128]. The metabolic shift in immune cells, often toward glycolysis and altered mitochondrial function, parallels the broader metabolic rewiring seen in bystander cells [95]. Inflammation propagation is further amplified by the activation of inflammasomes, such as NLRP3, which processes pro-IL-1β into its active form, fueling sustained cytokine release and pyroptotic cell death, a form of inflammatory cell death.

Complement system activation also contributes, with cleavage fragments (C3a, C5a) promoting immune cell chemotaxis, vascular permeability, and the amplification of inflammatory signaling pathways. In parallel, DNA-damaged or senescent bystander cells adopt the Senescence-Associated Secretory Phenotype (SASP), excreting a variety of inflammatory cytokines, proteases (e.g., MMPs), and growth factors that propagate tissue-level inflammation and remodeling [142,143,144]. SASP factors create a feedback loop that sustains chronic inflammation, with implications for fibrosis, tumor progression, and impaired regeneration [142,143]. In addition to that, the endothelial cells play an active role, with angiocrine signaling molecules (Angiopoietin-2, ICAM-1, E-selectin) facilitating immune cell adhesion and trafficking, bridging local injury with systemic immune responses [135,136,145,146,147]. Such interplay supports the emergence of abscopal effects, where immune modulation in irradiated tissue prompts anti-tumor or deleterious consequences at distant sites [136]. Thus, immune, and inflammatory pathways are central to both the propagation and extension of radiation BEs, integrating signals from damaged cells and modifying tissue fate in a temporally complex manner.

### 3.5. Stem Cell Niche, TME, and Systemic Effects

Radiation BEs signals profoundly impact the stem cell niches that sustain tissue regeneration and homeostasis. Non-irradiated stem and progenitor cells are influenced by paracrine factors like TGF-β, Notch ligands, and Wnt pathway modulators, leading to altered proliferation and differentiation [95,109,120,124,125,130,148,149]. For example, neural stem cells (NSCs) exposed indirectly to radiation modulate neurogenesis, while hematopoietic stem cells may undergo functional impairment or skewed lineage commitment [74,120,122,123,141,144,148,150,151,152,153,154,155,156,157,158,159,160,161,162,163,164]. The ECM undergoes dynamic remodeling during bystander signaling, involving matricellular proteins such as thrombospondin, osteopontin, and tenascin-C [153]. Matrix metalloproteinases (MMP2, MMP9) and ECM-degrading enzymes reshape the tissue architecture, affecting cell adhesion, migration, and mechanical cues [154,155]. These biomechanical alterations feed back into cellular mechanotransduction pathways (YAP/TAZ), influencing transcriptional programs and reinforcing bystander signaling [156]. Endothelial cells modulate the microenvironment by releasing angiocrine factors that regulate vascular permeability and immune cell infiltration [145,146,147]. Changes in the microvasculature can promote both regeneration and inflammation, underscoring the dual role of the tissue microenvironment in balancing damage and repair [128,134].

Strikingly, these local events ripple outward to systemic levels. Radiation-induced release of circulating cytokines, metabolites, and stress hormones activates the hypothalamic–pituitary–adrenal axis and can influence distant organs via neuroendocrine signaling. The abscopal effects exemplifies this phenomenon, where localized irradiation induces responses, including DNA damage and immune changes, in non-irradiated sites such as the heart, brain, or bone marrow [91,103,125,136,139,157]. Moreover, emerging evidence reveals that microbiome composition and metabolic products, including short-chain fatty acids and microbial ROS/NO, modulate BEs via immune and metabolic pathways [150,158,159]. Additionally, circadian rhythm disruption after radiation influences DNA repair timing and metabolic cycles, while age and sex hormones further modulate the susceptibility and nature of bystander responses [160]. Thus, radiation BEs transcends individual cells and tissues, orchestrating complex systemic adaptations that hinge on the intimate crosstalk between stem cells, ECM, vasculature, immune system, and organismal physiology.

### 3.6. Unique and Emerging Mechanisms

Beyond classical pathways, a suite of recently recognized mechanisms enriches the complexity of BEs, highlighting novel biological principles. Mechanotransduction pathways, involving mechanosensitive Piezo channels and YAP/TAZ transcriptional regulators, translate altered tissue stiffness and physical stress induced by radiation into biochemical signals [156]. These mechanotransduction signals integrate with biochemical inputs to adapt cellular behavior and influence bystander communication [161,162]. Unfolded protein response (UPR) and endoplasmic reticulum (ER) stress sensors (PERK, ATF6, IRE1) are activated in irradiated and bystander cells, influencing apoptotic thresholds, inflammation, and proteostasis [161,162]. Perturbations in the ubiquitin-proteasome system further impact protein turnover and stress responses [163,165]. Ion channel remodeling alters cellular excitability and intracellular calcium signaling, affecting processes like apoptosis and cell motility [166,167]. RNA-level regulation is expanded by A-to-I editing and diverse non-coding RNA species, tuning stress and immune gene expression finely [111,133]. Additionally, lipid rafts and membrane organization modulate receptor clustering and vesicle trafficking, contributing to propagation fidelity and signal amplification [168,169,170]. LLPS phenomena affect biomolecular condensates’ formation and dissolution under radiation stress, impacting RNA processing and transcription factor activity.

### 3.7. Mechanistic Diversity in Radiation BEs

While the diversity of radiation BEs molecular mechanisms is broad, we should differentiate between firmly established pathways with strong experimental evidence and those that are mainly speculative or require additional confirmation [44,80]. This distinction is necessary to narrow the gap for translating laboratory observations into clinically useful interventions and to adequately assess the therapeutic implications of BEs [44,80].

Widely established processes with reproducibility in multiple independent laboratories and experimental models are as follows: (1) Gap junction-mediated intercellular communication, namely through connexin43 channels, consistently demonstrated in a variety of cell types and radiation modalities [80,171]. GJIC involvement in the propagation of calcium ions, ROS, and small signaling molecules is supported by reproducible pharmacological blocking experiments by gap junction blockers [80,171]. (2) Signaling of soluble factors, especially release and activity of pro-inflammatory cytokines (TNF-α, IL-6, TGF-β) and ROS/RNS, is the best-characterized pathway [80,171]. Medium transfer experiments consistently showed that conditioned media from irradiated cells can induce DNA damage, micronuclei induction, and cell death in naive recipient cells [80,171]. (3) NFκB-dependent signal cascades have been broadly established as principal BEs mediators, with replicable data for NFκB activation caused by radiation leading to cytokine induction and bystander signal transmission [80,171]. Both genetic intervention and pharmacological inhibition across many model systems offer evidence for the pathway’s contribution.

Moderately established mechanisms with significant but not comprehensive supporting evidence are as follows: (1) Signaling through extracellular vesicles, though appealing in concept and having supporting evidence from a series of experiments showing the delivery of exosomal miRNA and proteins, remains inconsistent in experimental findings and lacks standardized protocols for isolation and characterization of vesicles [172]. (2) ROS amplification and mitochondrial dysfunction pathways show consistent involvement but with considerable heterogeneity of magnitude and duration of effects between and within cell types and different experimental conditions [172]. (3) p53 and activation of the DNA damage response in bystander cells is firmly established but displays context-dependent heterogeneity, particularly concerning threshold doses for activation and the duration of responses [172].

Speculative or novel mechanisms with substantial additional evidence required are as follows: (1) Mechanisms of cell-free chromatin (cfCh) transfer and integration, while acceptable by virtue of new experimental approaches, are controversial because of the lack of independent confirmation and questions surrounding the biological viability of stable genomic integration [117,173]. (2) Communication via tunneling nanotubes is an attractive concept with initial evidence, but requires more robust evidence of their prevalence and functional significance in radiation response [117]. (3) Liquid–liquid phase separation (LLPS) and mechanotransduction pathways (Piezo channels, YAP/TAZ signaling) are new ideas with limited proof related to radiation [173]. (4) Trained immunity and epigenetic bystander effect transmission, while supported by new evidence, require long-term experiments and mechanistic evidence to establish causalities [117].

Main limitations of current mechanistic information are as follows: (1) Limitations of experimental models: The majority of mechanistic data result from cell culture in vitro experiments, which may not reflect the complex tissue architecture, immunological interactions, and systemic effects present in living organisms [174]. (2) dose–response heterogeneity: Many mechanisms exhibit non-uniform dose–response relations across studies, some exhibiting “all-or-nothing” responses while others exhibit linearity or thresholds [175,176]. (3) Species and tissue specificity: Established mechanisms in one model system (e.g., human fibroblasts) prove to be variably reproducible in another cell type or species, and hence generalizability is poor [69]. (4) Temporal dynamics uncertainty: While acute bystander effects are well defined, the long-term maintenance and transgenerational inheritance of these effects remain poorly understood [177].

Clinical translation implications: The heterogeneity of mechanistic data demands cautious interpretation when making BEs-targeted interventions [44,178]. Therapeutic intervention should target known pathways (gap junctions, soluble factors, NFκB signaling) but accept that new mechanisms may present future opportunities for more selective modulation [179]. Predictive biomarker and personalized countermeasure development should also accommodate the known variation in susceptibility to BEs between people and tissues [58].

Building on these emerging insights, it becomes increasingly important to distinguish between well-established BEs pathways, supported by robust experimental evidence, and those that remain largely speculative or require further validation. Clarifying this distinction not only sharpens our mechanistic understanding but also guides the translation of laboratory findings into clinically actionable strategies, informing both therapeutic radiation design and protective interventions against unintended tissue effects.

### 3.8. Rationale and Impact of In Vitro Findings

In vitro experiments have been at the center of explaining the mechanistic bases of radiation BEs, offering controlled systems to investigate intercellular communication on a molecular level [86,180]. Although they cannot imitate the architectural complexity, immune contributions, or stromal environment of living tissues, such models remain crucial to establishing causal relationships [43]. For example, microbeam irradiation experiments have shown that focused irradiation of isolated cells can initiate genomic instability and micronucleation in neighboring non-irradiated cells [181]. Medium transfer assays further showed that soluble signals emitted from irradiated cultures are able to cause DNA damage and epigenetic changes in naïve recipients and that NTEs can occur irrespective of direct radiation tracks [182,183]. Evidently, the in vitro approaches not only provided the first mechanistic insights, but also laid the foundation to identify genetic and molecular determinants, biomarker design and/or the development of targeted countermeasures [184].

### 3.9. Prioritization and Translational Value of Animal and Human Studies

To give higher priority to the clinical relevance of BEs, the evidence based on animal models and human studies must be preferentially selected since these models better reflect the physiological context and complexity of disease [185]. Partial-body irradiation and local shielding experiments in vivo have provided evidence that intercellular communication and epigenetic alterations extend beyond irradiated fields to produce systemic responses like aberrant DNA methylation, chronic inflammation, and tissue remodeling in shielded organs [186]. Translational information from patient cohorts—like pediatric survivors and conformal RT-treated patients—offer proof that BEs contribute to secondary cancer, neurocognitive impairment, and off-target toxicities, substantiating the clinical demand for detecting and managing BEs in modern RT [187].

### 3.10. Radiation Quality and Radiation BEs

The characteristics and severity of BEs are highly dependent on the quality of the radiation, particularly its type and linear energy transfer (LET) [188]. Comparison experiments demonstrate that high-LET radiation, such as alpha particles and heavy ions (e.g., carbon or iron ions), deposits energy densely along its path, causing clustered DNA damage, including double-strand breaks and complex lesions, and persistent genomic instability in bystander cells that is difficult to repair [189,190]. These irradiated cells release a cascade of signaling molecules, such as ROS, NO, cytokines like TGF-β and IL-6, and extracellular vesicles containing microRNAs and damaged DNA fragments, that propagate stress responses to neighboring cells. These bystander cells may then undergo oxidative stress, DNA damage, apoptosis, senescence, or even malignant transformation, despite not being directly irradiated. These complex events imply that high-LET modalities (now being applied in particle therapy and FLASH RT, etc.) require distinct risk-benefit assessment and possibly discrete countermeasures [43]. In contrast, low-LET radiation such as γ-rays and X-rays distributes energy more diffusely, resulting in less severe DNA damage and a more variable bystander response. The magnitude of BEs in low-LET contexts is often modulated by factors such as radiation dose, dose rate, cell type, and genetic background, particularly the status of tumor suppressor genes like p53, which influence DNA damage response pathways [190]. Importantly, the persistence and systemic nature of BEs, especially following high-LET exposure, have profound implications for RT, where non-targeted tissue damage may contribute to side effects, and for space exploration, where astronauts are exposed to high-LET cosmic radiation. Understanding the differential impact of radiation quality on BEs is therefore essential for optimizing therapeutic strategies, improving radioprotection protocols, and assessing long-term risks associated with radiation exposure [191].

### 3.11. Dose-Rate Modulation of Radiation BEs

Dose-rate is a critical parameter influencing radiation BEs [44]. Conventional dose-rate exposures allow for continuous generation of reactive oxygen and nitrogen species (ROS/RNS) and cytokine signaling, facilitating the propagation of bystander effects through intercellular communication and paracrine mechanisms [192]. In contrast, ultra-high dose-rate irradiation, such as in FLASH radiotherapy (RT), establishes a distinctive physicochemical environment, including transient oxygen deficiency and altered radical chemistry, which may impede classical bystander signaling cascades [193]. Experimental evidence indicates that these conditions can also influence subsequent events such as DNA damage accumulation, inflammatory signaling, and epigenetic alterations within non-irradiated cells [44]. Recent studies further demonstrate that particle LET and fluence affect the complexity and frequency of clustered DNA damage, suggesting that high-LET particles can induce more complex DNA lesions, which may impact bystander responses [180,194]. Collectively, these findings indicate that dose-rate, along with LET and fluence, shapes the extent and nature of bystander signaling, with implications for DNA damage propagation, inflammatory pathways, and epigenetic regulation (Table 2).

## 4. Clinical Implications and Significance of Radiation BEs

Radiation BEs have emerged as a clinically significant phenomenon that expands the conventional boundaries of radiation biology and medicine including cancer RT. Traditionally, RT has been understood in terms of direct DNA damage within irradiated tissues. However, research over the past two decades has clarified that non-irradiated cells, both within and beyond the radiation field, can elicit biological responses secondary to signals released by directly exposed cells [39]. These responses carry broad implications for oncologic outcomes, normal tissue toxicity, therapeutic planning, and survivorship. Radiation BEs manifest with diverse clinical consequences across multiple organ systems, often amplifying both acute and late toxicities beyond directly irradiated volumes [121]. Among these, CNS toxicity provides a striking example of how low-dose exposure and systemic signaling interplay to produce functional deficits [234]. Clinical studies of cranial RT in pediatric and adult patients have repeatedly observed neurocognitive decline, even when critical structures such as the hippocampus and contralateral brain hemispheres receive only scattered doses [235,236]. Hippocampal-sparing IMRT has been proposed to mitigate these effects, yet retrospective analyses reveal persistent memory and processing speed deficits in cohorts treated with this approach, suggesting additional non-local mechanisms at play [237]. Preclinical models reinforce this phenomenon: irradiated cortical neurons release pro-inflammatory cytokines (IL-1β, TNF-α) and EVs carrying miR-1246 and miR-21, which induce microglial activation and synaptic remodeling in non-irradiated regions [238,239]. MRI studies of medulloblastoma survivors demonstrate diffuse white matter changes and reduced cortical thickness years after treatment, correlating with impaired attention and executive function [240,241]. These findings emphasize the need to address BEs-mediated neuroinflammation in neuroprotective strategies [242].

Cardiac toxicity similarly exhibits signatures of radiation BEs [243]. While direct myocardial irradiation explains part of the observed ischemic and fibrotic changes, data from breast cancer and Hodgkins lymphoma survivors implicate systemic bystander signaling in out-of-field cardiac damage [244,245]. Studies have reported a linear increase in major coronary events of 7.4% per Gy mean heart dose, with detectable risks at doses as low as 2Gy [244]. RT has been implicated in the development of constrictive pericarditis, valvular heart disease, coronary artery disease (CAD), conduction abnormalities, and heart failure [246,247]. The incidence of RT-induced cardiovascular disease (CVD) in surviving cancer patients is estimated to be between 10% and 39% by 10 years post-treatment, with up to 88% of patients demonstrating asymptomatic cardiac abnormalities. Clinically evident heart failure after RT, is primarily manifested as left ventricular diastolic dysfunction, as opposed to the systolic dysfunction with chemotherapy [246,248]. On the other hand, premature CAD is a well-recognized complication of RT, with a reported cumulative incidence of 21% at 20 years in individuals treated with RT in the presence of other cardiovascular risk factors such as obesity, hypertension, and dyslipidemia [246]. Evidently, the diagnosis of CAD following RT in young patients who have minimal risk factors for arterial sclerosis [249] is a compelling evidence for a causal relationship between RT and CVD. Second, vascular damage, in general, is a known risk factor for accelerated atherosclerosis and cardiovascular events [250]. However, it still remains uncertain how vascular damage pre-dispose individuals to heightened risk of CVD. To that end, subclinical myocardial strain and endothelial dysfunction have been documented even in contralateral breast cancer patients receiving negligible cardiac doses, suggesting contributions from inflammatory cytokines (TGF-β1, IL-6) and ROS generated in irradiated fields [251,252]. Animal models support this hypothesis, showing that conditioned media from irradiated endothelial cells induce apoptosis and senescence in non-irradiated cardiomyocytes [253]. Clinically, biomarkers such as high-sensitivity troponins, NT-proBNP, and endothelial microparticles are being explored to monitor early BEs-related cardiac injury, with implications for long-term surveillance and intervention [254,255].

Pulmonary complications, particularly radiation pneumonitis and fibrosis, frequently extend beyond high-dose regions, reflecting both direct damage and propagated radiation BEs [256,257]. Meta-analyses of lung cancer cohorts treated with IMRT or VMAT reveal pneumonitis in low-dose regions (V5-V10) that exceed predictions based on dose-volume histograms (DVHs) [258,259]. Prospective studies have identified surges in serum TGF-β1 and IL-8 levels during RT as predictors of grade 2 or higher pneumonitis, consistent with systemic inflammatory amplification [260,261]. Preclinical investigations demonstrate that irradiated lung epithelial cells release exosomes enriched in pro-fibrotic miRNAs (miR-21, miR-155), which stimulate fibroblast proliferation and collagen deposition in adjacent and distal lung tissue [262,263]. Clinically, antifibrotic agents such as pirfenidone and nintedanib, approved for idiopathic pulmonary fibrosis, are being investigated for their potential to modulate BEs-mediated lung injury [264,265].

In the gastrointestinal (GI) system, radiation BEs contribute to both acute and chronic toxicity [266]. Pelvic RT for prostate, rectal, and gynecologic cancers is associated with enteritis, proctitis, and late fibrotic changes that often extend beyond the planned target volume [267]. Patients receiving extended-field RT demonstrate altered gut microbiota composition, increased intestinal permeability, and persistent inflammatory infiltration even in unirradiated bowel segments [268]. Preclinical studies implicate irradiated enterocytes in releasing DAMPs and pro-inflammatory cytokines, which recruit immune cells and amplify mucosal injury [269,270]. Therapeutic strategies, including fecal microbiota transplantation (FMT) and probiotic supplementation, are under investigation to restore gut homeostasis and mitigate BEs-driven enteropathy [271].

Cutaneous toxicity, while often attributed to direct irradiation, also shows features consistent with bystander phenomena [272,273]. Radiation dermatitis, fibrosis, and telangiectasia occasionally extend into skin regions receiving only scattered doses [274]. Histopathological studies of irradiated breast cancer patients reveal myofibroblast activation and extracellular matrix remodeling in skin beyond the primary field [275,276,277]. These findings are mirrored in animal models where non-irradiated dermal fibroblasts exhibit increased expression of TGF-β1 and α-SMA upon exposure to conditioned media from irradiated keratinocytes [278,279]. Clinical trials of pentoxifylline combined with vitamin E have demonstrated partial reversal of chronic radiation-induced fibrosis, possibly through modulation of these bystander-mediated pathways [280,281].

Secondary malignancies represent one of the most feared late effects of RT, particularly in young patients whose long-life expectancy magnifies cumulative risks [282,283,284]. While direct DNA damage in irradiated tissues remains a primary driver, accumulating evidence points to the critical role of radiation BEs in propagating RIGI and malignant transformation in non-irradiated tissues [44,285,286,287]. Epidemiologic data from large survivor cohorts, such as the Childhood Cancer Survivor Study (CCSS) and British Childhood Cancer Survivor Study (BCCSS), have documented significant increases in second cancers occurring outside irradiated fields [283,284,288]. For instance, CCSS reported a relative risk of 3.6 for secondary malignancies in tissues receiving doses ≤ 5 Gy, implicating systemic mechanisms independent of scatter or leakage radiation [284,289,290]. Similarly, survivors of pediatric Hodgkin lymphoma demonstrate elevated rates of breast, thyroid, and gastrointestinal cancers decades after mantle field irradiation, with patterns poorly predicted by direct dose distributions alone [289,291,292].

Mechanistically, preclinical studies provide compelling support for BEs-mediated secondary malignancies [64]. In vitro experiments show that medium transfer from irradiated fibroblasts induces chromosomal aberrations, micronuclei formation, and persistent γ-H2AX foci in naïve recipient cells, hallmarks of RIGI [53,270]. Animal studies extend these findings, revealing increased tumor incidence in shielded body regions following partial-body irradiation [98,150]. These phenomena are linked to soluble factors such as TGF-β1 and EVs carrying oncogenic miRNAs (miR-21, miR-222) and altered DNA methylation profiles, creating a pro-tumorigenic microenvironment [69,292,293]. Clinically, this underscores the need for lifelong vigilance and tailored surveillance protocols, especially in pediatric survivors who often received extended-field RT during critical developmental windows [294,295].

The pediatric population exhibits unique vulnerabilities to radiation BEs due to rapidly proliferating tissues, immature DNA repair machinery, and dynamic epigenetic landscapes [296,297]. Neurodevelopmental consequences of cranial irradiation in children illustrate the interplay between direct and radiation BEs [298,299]. Cognitive decline, reduced IQ, and psychosocial impairments have been observed even in survivors treated with conformal techniques that spared the hippocampus [300,301]. Studies implicate bystander-mediated neuroinflammation, with activated microglia and astrocytes releasing cytokines (IL-1β, TNF-α) and ROS, propagating damage throughout the CNS [302,303]. Longitudinal imaging in these patients often shows progressive white matter loss and cortical thinning, correlating with functional impairments [304,305]. Importantly, the impact of radiation BEs may extend into adulthood, predisposing survivors to neurodegenerative diseases, cardiovascular complications, and metabolic syndrome.

Together, the recognition of BEs as a clinically significant phenomenon challenges the traditional method of RT as a strictly localized intervention [58,306]. From personalized treatment planning and pharmacologic modulation to advanced delivery technologies and survivorship care, integrating radiation BEs into clinical oncology represents a necessary evolution [307]. As translational research continues to elucidate the molecular underpinnings of BEs, the field stands poised to redefine RT toxicity models and expand therapeutic windows, ultimately improving both the quantity and quality of life for cancer survivors [308,309].

## 5. Countermeasures for Radiation BEs

As the biological relevance of radiation BEs has become apparent in preclinical, bed-bench and clinical settings, focus has increasingly turned toward countermeasures that could be implemented during or after clinical RT. The following section surveys the current state of such interventions, highlighting both pharmacologic agents and biophysical strategies that show promise in mitigating BEs-driven toxicity (Figure 2).

Pharmacological countermeasures offer a direct means of disrupting the molecular signals underpinning bystander-mediated damage [310]. By targeting key pathways such as oxidative stress [310], cytokine signaling [311], and intercellular communication [66], these agents form a foundational component of emerging radioprotective strategies (Table 3).

### 5.1. Cyclooxygenase-2 (COX-2) Inhibitors

COX-2 and its downstream prostaglandins have been repeatedly implicated in the propagation of radiation BEs, serving as mediators of pro-inflammatory cytokine release and oxidative stress [312]. Multiple preclinical studies have demonstrated the efficacy of COX-2 inhibitors in suppressing bystander responses [85]. Studies have shown that NS-398, a selective COX-2 inhibitor, significantly reduced micronucleus formation in unirradiated cells co-cultured with irradiated fibroblasts [85]. Animal studies demonstrate that COX-2 inhibitors like indomethacin and celecoxib can reduce PGE_2_-mediated inflammatory damage in irradiated salivary glands and normal tissues [313]. However, direct evidence of reduction in γH2AX foci or other bystander damage markers in shielded, non-irradiated tissues is not yet reported in the literature. Importantly, these effects have been observed in both animal models and in human primary cell co-culture systems, supporting translational potential [314]. Together, prolonged COX-2 inhibition carries risks of gastrointestinal and cardiovascular toxicity, and suppression of prostaglandin-mediated tissue repair could counteract potential protective effects in some contexts [315].

### 5.2. Antioxidant Therapies

Oxidative stress plays a central role in radiation BEs, with ROS serving as both initiators and propagators of damage [133]. Antioxidants have therefore been widely studied as potential mitigators [311,312]. Early work demonstrated that administration of N-acetylcysteine (NAC) to mice exposed to total body irradiation attenuated oxidative stress markers and chromosomal aberrations in shielded tissues [313]. Vitamin E analogs, such as α-tocopherol and synthetic compounds like tempol, also reduced DNA damage and inflammatory responses in unirradiated bystander cells [314]. In human studies, antioxidant supplementation before RT has shown mixed results, with some trials reporting reduced normal tissue toxicity, but concerns about tumor protection remain unresolved [311,313]. A notable finding indicated the upregulation of endogenous antioxidant enzymes (e.g., SOD, catalase) in distant organs after partial-body irradiation, suggesting a natural adaptive response that exogenous antioxidants could amplify [315].

### 5.3. Melatonin and Related Compounds

Melatonin has emerged as a potent radioprotective agent due to its antioxidant, anti-inflammatory, and mitochondrial regulatory properties [316,317,318]. Studies demonstrated that melatonin pre-treatment reduced oxidative DNA damage and lipid peroxidation in both irradiated and non-irradiated tissues [319,320]. In a rodent model, melatonin administered at 10 mg/kg intraperitoneally before partial-body irradiation significantly reduced levels of IL-6 and TNFα in distant, shielded organs [319]. Its ability to penetrate cellular and nuclear membranes and scavenge a broad spectrum of free radicals underscores its potential as a systemic BEs countermeasure [321].

### 5.4. Statins, Metformin, and Herbal Extracts

Statins, traditionally used for lipid lowering, have shown promise as radioprotectors by modulating endothelial function and reducing inflammatory signaling [322,323]. Lovastatin, for example, suppressed BEs-associated micronuclei in vitro through inhibition of Rho signaling [324]. Metformin, an AMPK activator, has demonstrated ROS-lowering and DNA damage-reducing effects in irradiated and bystander cells, raising interest in its dual role as an anti-cancer adjuvant and radiation BEs mitigator [325,326]. Natural compounds such as curcumin, resveratrol, and green tea polyphenols have also been investigated, with studies showing reductions in oxidative stress and cytokine release in bystander models [101,183,327,328].

To effectively translate this growing understanding into clinical practice, considerable effort has been directed toward identifying and modulating specific molecular targets involved in bystander signaling, paving the way for the development of experimental interventions to mitigate radiation BEs.

### 5.5. Cell-Free Chromatin (cfCh) Degradation Strategies

Liquid biopsies represent another promising avenue for real-time monitoring of systemic radiation responses [328,329]. These nucleosomal DNA-histone complexes can enter neighboring cells, integrate into their genomes, and trigger DNA damage responses and inflammatory signaling [117,183]. In culture models, BrdU-labeled cfCh from apoptotic cells enters bystander nuclei and induces γH2AX foci, NFκB activation, IL-6 secretion, and chromosomal aberrations [117,183]. Crucially, interventions that dismantle or neutralize cfCh suppress these BEs [117]. DNase I, added to conditioned medium, cleaves cfCh DNA and renders the particles biologically inert [117,183]. Nanoparticle-bound anti-histone antibodies (chromatin-neutralizing nanoparticles, CNPs) bind cfCh histones and promote their clearance, blocking nuclear uptake and downstream γH2AX and NFκB induction [117,173,183]. Both approaches substantially reduced labeled cfCh uptake in co-culture assays [183]. Resveratrol-copper (R-Cu) complexes offer a third strategy, oxidatively fragmenting extracellular DNA without harming host genomes [183,329,330]. Low-dose resveratrol combined with trace copper generates ROS that selectively degrade cfCh [183,331]. In murine models, oral R-Cu increased brain ROS levels yet reduced γH2AX staining, consistent with extracellular cfCh degradation [183,332]. Notably, host cell DNA remained unaffected [183,331]. Although these methods differ, they share a common goal: dismantling cfCh before it initiates genotoxic and inflammatory cascades [117,183]. Preclinical studies provide proof-of-concept for enzymatic cleavage, antibody-mediated neutralization, and oxidative fragmentation [117,183,330,332]. Translation to humans, however, faces challenges [320]. Systemic DNase I (e.g., Pulmozyme) could have off-target effects on host DNA [333]. CNPs are experimental and may trigger immune responses [183,334]. R-Cu, using widely available agents, appears safe at low doses in small clinical studies but remains untested for BEs [183,335,336].

### 5.6. Targeting microRNAs and Exosome-Mediated Signaling

IR reprograms the secretome of irradiated cells, with exosomes and their microRNA (miRNA) cargo emerging as potent mediators of BEs [122,336,337]. Exosomes released post-irradiation have been shown to induce DNA damage and stress markers in unexposed bystanders, implicating RNA-mediated intercellular communication [122,338]. Among these, miR-21 stands out as a key effector [74]. Irradiation upregulates miR-21 in donor cells, which is packaged into exosomes and transferred to neighboring cells [74,122]. In recipient bystanders, exosomal miR-21 downregulates anti-apoptotic targets such as Bcl-2 and activates DNA damage signaling, propagating BEs [74,122]. Preemptive inhibition of miR-21 using antagomirs significantly blunts these effects, directly linking this miRNA to bystander genotoxicity [122,338]. Conversely, some miRNAs may mitigate bystander responses. miR-146a, a well-characterized anti-inflammatory regulator, suppresses NFκB signaling and cytokine production, including TNF-α, IL-6, and other SASP factors [339]. In certain bystander models, miR-146a is modestly downregulated in irradiated environments, suggesting its loss may contribute to heightened inflammation [340]. Strategies to restore or enhance miR-146a delivery, for example, through exosomes enriched in this miRNA, could hypothetically temper BEs-associated cytokine cascades [341]. Though direct evidence in BEs is limited, its anti-inflammatory role in immune contexts supports this concept [342]. More broadly, disrupting exosome pathways can limit miRNA transfer [107]. The neutral sphingomyelinase inhibitor GW4869 blocks ceramide-dependent multivesicular body formation and thereby reduces exosome biogenesis [343]. In cancer models, GW4869 markedly lowers exosomal miR secretion, though it remains untested in radiation BEs [343,344]. Additional approaches, such as inhibiting exosome uptake (e.g., heparin, annexin V) or degrading extracellular miRNAs with RNases, could also suppress bystander signaling [345]. However, these interventions are nonspecific and lack clinical approval [345]. Collectively, these findings highlight exosomal miR-21 as a prototypical pro-BEs signal, while miR-146a and related anti-inflammatory miRNAs (e.g., miR-223, miR-181c) may counteract cytokine-driven BEs [74,346]. Targeting production, release, or uptake of exosomes offers a conceptual route to dampen BEs, but translation to humans will require solutions to specificity, delivery, and toxicity challenges.

### 5.7. Mitochondrial and Metabolic Regulators

The Mitochondria act as central hubs in radiation BEs signaling by generating danger signals and sustaining energy metabolism [224]. Irradiated cells release ATP and nucleotides (ADP, UTP) into the extracellular space through connexin hemichannels and pannexins [346,347]. These nucleotides activate purinergic P2 receptors on neighboring cells, modulating calcium flux, MAPK, and DNA repair pathways [210,348]. Studies proposed that extracellular ATP/UTP/UDP drive TRPM2-mediated calcium influx and P2X7 activation in donor cells, further amplifying ATP release [210,349]. This positive feedback stimulates P2Y6/12 on bystanders, engaging EGFR-ERK and antioxidant responses [210,350]. Pharmacologic depletion of ATP with apyrase or blockade of P2 receptors (A438079, suramin) theoretically dampens these effects, though BEs-specific validation is lacking [112,348,350]. Similarly, carbenoxolone inhibition of ATP-permeable connexins might impede signal transmission [351]. Mitochondrial ROS also play a critical role. Irradiated cells exhibit increased superoxide and H_2_O_2_ production, which propagate oxidative stress in bystanders [352]. Antioxidants like N-acetylcysteine and mitochondrial-targeted MitoQ applied to donor cultures abrogate bystander γH2AX foci and apoptosis, underscoring ROS-driven mechanisms [352,353]. Likewise, mitochondrial uncouplers or inhibitors (rotenone, antimycin A, oligomycin) suppress bystander DNA damage when administered during irradiation [108]. Studies in ρ^0^ lymphoblasts further demonstrate that mitochondrial integrity is essential for both generating and sensing bystander signals [270]. However, these agents are cytotoxic, and only transient inhibition during irradiation appears tolerable in preclinical models [224]. On the metabolic side, glycolytic flux, and NAD^+^/NADH balance may influence radiation BEs [354]. Irradiated cells often upregulate glycolysis and secrete lactate, providing metabolic fuel for bystanders [355]. Interventions like 2-deoxyglucose could theoretically disrupt this metabolic coupling, but direct evidence is sparse [356]. More promising are modulators of mitochondrial metabolism, such as metformin or phenformin, which inhibit complex I and activate AMPK, indirectly boosting antioxidant defenses [357]. While proposed as radioprotectors, their impact on radiation BEs remains unexplored [357,358]. In summary, mitochondrial ATP release and ROS production propagate BEs through purinergic and oxidative mechanisms [224,347]. Experimental interventions targeting these pathways–antioxidants, P2 receptor antagonists, and mitochondrial inhibitors, demonstrate proof-of-concept in vitro [352,359]. However, clinical translation faces challenges of specificity, timing, and toxicity.

### 5.8. Inflammasome and Immune Targets

IR activates innate immune pathways, including inflammasomes, which may amplify BEs [89]. The NLRP3 inflammasome senses cellular stress signals such as ROS, mitochondrial DNA, and potassium efflux, leading to caspase-1 activation [218]. Caspase-1 then processes pro-IL-1β into its mature form and induces pyroptosis through gasdermin D cleavage [360]. In irradiated bone marrow macrophages, doses as low as 5-10 Gy induce caspase-1 cleavage, LDH release, and secretion of IL-1β and IL-18 in a dose-dependent manner [221]. These findings suggest that inflammasome activation contributes to radiation-induced inflammation and may also affect bystander cells exposed to secondary signals like extracellular DNA or ATP, canonical activators of AIM2 and NLRP3 [221,361,362]. Targeting inflammasome pathways is a promising mitigation strategy. NLRP3 inhibitors (e.g., MCC950) and caspase-1 inhibitors (VX-765, Belnacasan) have shown efficacy in reducing IL-1β secretion and attenuating radiation-induced tissue inflammation in animal models [363,364]. Clinically approved agents such as anakinra (IL-1 receptor antagonist) could potentially interrupt a BEs feedback loop by neutralizing IL-1β released from irradiated or senescent cells [365]. Similarly, disulfiram blocks gasdermin pores, preventing pyroptotic lysis and cytokine release, while antioxidants like NAC and SOD mimetics indirectly suppress inflammasome priming by scavenging ROS [352,366,367]. Although no BEs-specific studies have yet tested these interventions, mechanistic evidence supports their potential [352,367]. Importantly, transient administration around RT may minimize immunosuppression risks associated with chronic IL-1 blockade [365]. The ability of IL-1β inhibitors to reduce radiation-induced fibrosis in preclinical models further supports this strategy [368]. In summary, the inflammasome-IL-1β axis amplifies radiation injury and likely contributes to radiation BEs [364]. Pharmacologic inhibitors of NLRP3, caspase-1, gasdermin D, or IL-1 signaling are promising candidates to dampen inflammatory bystander effects, warranting further investigation [364].

### 5.9. Senescence and SASP-Mediated Bystander Signaling

Persistent DNA damage and oxidative stress from IR drive cells into senescence, a state of permanent cell-cycle arrest marked by continued metabolic activity and secretion of the senescence-associated secretory phenotype (SASP) [369]. This cocktail of cytokines (IL-1α/β, IL-6, IL-8), growth factors (VEGF), and proteases creates a pro-inflammatory microenvironment that propagates damage to neighboring cells [143]. SASP factors promote inflammation, secondary senescence, and tissue dysfunction, contributing to late effects such as fibrosis, impaired regeneration, and possibly secondary malignancies [105,110]. Pharmacologic strategies to mitigate SASP-driven bystander effects fall into two categories: senolytics and senostatics [369]. Senolytics, such as dasatinib plus quercetin (D+Q), navitoclax (ABT-263), and fisetin, selectively eliminate senescent cells by targeting their survival pathways [370,371]. In murine models, D+Q reversed radiation-induced muscle weakness and improved vascular parameters [369], while navitoclax reduced pulmonary fibrosis and restored hematopoietic stem cell function by clearing senescent fibroblasts and bone marrow cells [372,373]. Senostatics, in contrast, suppress SASP production without killing senescent cells [143]. Agents like rapamycin (mTOR inhibitor) and metformin (AMPK activator) inhibit NFκB-driven cytokine secretion and reduce senescent cell accumulation [358,374]. Short-term administration of these drugs after irradiation has lowered senescence markers in mouse tissues [373]. Their existing FDA approval offers a translational advantage, and ongoing trials are exploring their potential as radioprotectors [370,373]. Timing and specificity remain critical considerations [373]. Interventions must target senescence once established to avoid disrupting beneficial acute stress responses [369,373]. Most evidence comes from animal studies on normal tissue late effects, and human relevance remains under investigation. Safety concerns, such as navitoclax-induced thrombocytopenia, underscore the need for carefully timed, intermittent dosing regimens to minimize toxicity [375,376]. In summary, radiation-induced senescence contributes to chronic bystander signaling via the SASP [369,373]. Preclinical studies support the use of senolytic and senostatic agents to blunt this process, but their application to radiation BEs in humans requires further validation.

### 5.10. Precision and Personalized Countermeasures

The emergence of precision oncology has provided a fertile ground for incorporating the biological complexity of BEs into patient-specific treatment planning and toxicity mitigation strategies [377]. Advances in genomics, transcriptomics, and proteomics have revealed significant interindividual variability in radiation BEs’ susceptibility, driven by germline genetic variations, epigenetic states, and immune competence [378,379]. Single-nucleotide polymorphisms (SNPs) in key DNA repair genes (ATM, BRCA1, XRCC1), antioxidant pathways (SOD2, GPX1), and cytokine signaling molecules (IL6, TNFA) have been implicated in modifying the extent and persistence of bystander responses [380,381]. For instance, the ATM rs1801516 polymorphism has been associated with increased late radiation toxicity, potentially exacerbated by RIGI in unirradiated tissues [382,383]. Similarly, SOD2 polymorphisms modulate the ability of cells to quench ROS, thereby influencing the propagation of oxidative stress beyond irradiated volumes [384].

### 5.11. Emerging Translational Approaches

Pharmacologic modulation of radiation BEs has emerged as a translational priority, with several agents advancing through preclinical and clinical pipelines [385,386]. Radioprotectors such as amifostine, long known for its thiol-based free radical scavenging properties, have shown limited uptake due to toxicity and inconsistent efficacy [387]. More selective agents are now under investigation [337]. Avasopasem manganese (GC4419), a superoxide dismutase (SOD) mimetic, demonstrated significant reductions in severe oral mucositis in a phase III trial (ROMAN) of head and neck cancer patients undergoing chemoradiotherapy [387,388]. By attenuating the amplification of ROS, Avasopasem manganese (GC4419) may help protect adjacent normal tissues from radiation-induced damage, potentially limiting harmful BEs [389,390]. IL-1 receptor antagonists such as anakinra are being explored to counteract pro-fibrotic and pro-inflammatory cascades driven by BEs, with early-phase trials evaluating their role in preventing radiation-induced pneumonitis and fibrosis [365,391]. Conversely, therapeutic strategies seek to amplify beneficial bystander responses in the tumor microenvironment to enhance systemic anti-tumor immunity [81,392]. The abscopal effect, whereby localized irradiation triggers regression of distant metastases, represents an archetypal beneficial BEs [393,394]. Trials combining SBRT with immune checkpoint inhibitors (ICIs) such as anti-PD-1/PD-L1 and anti-CTLA-4 antibodies (e.g., KEYNOTE-001, PACIFIC, NRG-GY017) have reported promising results across multiple cancer types [395,396,397]. Mechanistically, irradiated tumor cells release DAMPs and pro-inflammatory cytokines that recruit dendritic cells and cytotoxic T lymphocytes, a process that may be enhanced by modulating bystander signaling to favor immune activation over suppression [394,398]. DNA damage response (DDR) inhibitors such as PARP inhibitors (Olaparib) and ATR inhibitors (Ceralasertib) are also being evaluated for their ability to augment both direct and bystander-mediated tumor cell kill [399,400].

### 5.12. Radiopharmaceutical Therapy: High-LET Microdosimetry, Systemic Biodistribution, and BEs/Abscopal Biology

Most mechanistic data on BEs and NTEs derive from external-beam radiation setups, which deliver relatively uniform, acute, low-linear energy transfer (LET) doses to confined anatomical fields [401]. In contrast, radiopharmaceutical therapies (RPTs) deliver radiation randomized, with heterogeneous dose deposition across systemic compartments, prolonged exposure timelines, and mixed LET characteristics, warranting dedicated consideration of their unique bystander and abscopal signaling properties [401]. Beta-emitters such as ^177^Lu-PSMA-617 and ^131^I deliver low-LET radiation (~0.2 keV/μm) to micrometer-scale tissue clusters, inducing predominantly single-stranded DNA breaks [402,403]. Alpha-emitters including ^225^Ac and ^223^Ra dichloride, by contrast, produce highly localized high-LET tracks (50–230 keV/μm) within cell nuclei over short tissue ranges (50–100 μm), causing clustered double-stranded breaks difficult for cells to repair [404]. This radionuclide-dependent heterogeneity, coupled with selective tumor uptake, generates a distinctive microdosimetric pattern: very high absorbed doses to targeted malignant cells paired with low but biologically relevant doses to systemic host tissues via blood-pool clearance and off-target retention [405]. This dual-dose geometry creates opportunities for both local bystander signaling in the TME and systemic abscopal-type immune priming throughout the body [404].

High-LET damage in tumor cells preferentially triggers immunogenic cell death (ICD), amplifying release of DAMPs, including HMGB1, ATP, HSPs, and exosome secretion [401]. These mediators activate dendritic cells and CD8+ T cell responses, enhancing systemic immune priming more robustly than uniform low-LET exposure [406]. Concurrently, chronic low-dose exposure of bystander tissues via circulating radionuclides and long-lived decay products sustains inflammatory and adaptive signaling over protracted timescales (days to weeks), contrasting with the acute transient bystander responses typical of external-beam fields [401].

Preclinical evidence demonstrates that α-emitter RPT depletes regulatory T cells (Tregs) in TME while promoting CD8^+^ effector activation, for example, ^225^Ac-NM600 significantly reduced Treg infiltration and increased CD44^+^, CD69^+^, Ki67^+^, and CD8^+^ populations in murine prostate cancer models [407]. Clinical biomarker studies corroborate systemic immune activation: a Phase 2 trial of single-dose ^177^Lu-PSMA-617 with maintenance pembrolizumab showed that patients with objective responses exhibited significantly higher frequencies of circulating CD8^+^ effector memory cells and γδT cells compared to non-responders [408]. Additionally, ^223^Ra treatment increased CD8^+^ and CD4^+^ T cell proliferation in murine bone-metastatic prostate cancer models, and combination with anti-PD-1 antibodies significantly prolonged survival [409].

The protracted exposure timelines and LET heterogeneity of RPT complicate direct extrapolation of dose–response models derived from external-beam paradigms [401]. Accurate interpretation requires pharmacokinetic-informed dosimetry that captures organ retention and off-target biodistribution [401,404]. Countermeasures established for low-LET external beam exposures, such as global antioxidants, anti-inflammatory agents, or free radical scavengers, may be less effective against clustered high LET lesions or could interfere with desired immune priming. Thus, LET-specific preclinical testing of candidate countermeasures is necessary to ensure compatibility with RPT-induced immune effects [401,404].

This knowledge provides the foundation for rational combination strategies, such as integrating RPT with immune checkpoint inhibitors, and highlights the need for LET-tailored countermeasures that may differ fundamentally from those developed for low-LET, external-beam bystander suppression.

### 5.13. Survivorship and Long-Term Monitoring

Survivorship care represents a critical frontier for translating radiation BEs knowledge into improved long-term outcomes [309,376]. Persistent molecular disruptions, including DNA methylation, histone modifications, and sustained low-grade inflammation, extend the biological footprint of RT years beyond treatment cessation [69]. These processes contribute to late effects such as cardiovascular disease, neurocognitive decline, and secondary malignancies, necessitating proactive surveillance [410]. Biomarker panels incorporating C-reactive protein (CRP), high-sensitivity troponins, brain-derived neurotrophic factor (BDNF), and γH2AX foci are being explored to detect subclinical organ dysfunction [411,412]. Tailored interventions, such as statins and beta-blockers for cardio protection, cognitive rehabilitation programs, and lifestyle modifications, are under evaluation in survivorship trials aiming to mitigate BEs-driven morbidity [413,414].

**Preventing fire with fire**: While most BEs countermeasures target downstream mediators, procedural strategies modify how radiation is delivered to suppress the upstream cascades driving NTEs. Preclinical evidence suggests that innovations like ultra-high dose rate irradiation and spatial fractionation may not only reduce direct cytotoxicity but also attenuate systemic bystander signaling [38,414].

### 5.14. FLASH-RT Minimize Radiation BEs

FLASH RT represents a model shift in delivery strategies, characterized by ultra-high dose rates (>40 Gy/s) [415,416]. Ultra-high dose RT has garnered attention for its ability to spare normal tissues from radiation damage [38]. Beyond its capacity for reducing direct cytotoxicity, emerging data suggest that FLASH-RT also attenuates BEs [415,416]. Preclinical studies demonstrate that FLASH spares normal tissues from radiation-induced fibrosis, neurocognitive decline, and immunosuppression while maintaining tumoricidal efficacy [38,417]. Early-phase clinical trials are now evaluating their safety and effectiveness in murine models, with the potential to redefine RT toxicity profile with markedly lower levels of oxidative stress markers and pro-inflammatory cytokines (e.g., IL-6, TNF-α) compared to conventional dose-rate irradiation [418,419]. This dampened systemic response aligns with reduced secretion of bystander signals, including exosomes and mitochondrial ROS, from irradiated cells [270,419]. At the cellular level, the abbreviated delivery window of FLASH appears to mitigate mitochondrial dysfunction and minimize secondary ROS bursts, which are critical drivers of BEs [420]. In vitro studies using conditioned medium from FLASH-irradiated fibroblasts reveal significantly diminished γH2AX foci formation and micronuclei in recipient bystander cells relative to medium from conventional irradiations [421]. Additionally, the transient oxygen depletion observed during FLASH exposures may impair the oxidative processes that stabilize cfCh and pro-inflammatory lipid mediators [422,423]. These findings highlight the potential of FLASH-RT not only to reduce direct tissue injury but also to indirectly protect non-targeted tissues by suppressing bystander signaling pathways [415,423]. However, further mechanistic elucidation and clinical translation remain ongoing.

### 5.15. Advanced Particle Therapies

Recognizing these risks has spurred innovations in RT aimed at minimizing unintended tissue exposure and, by extension, bystander signaling [5,424]. PBT has gained prominence due to its ability to deposit energy at a defined depth (Bragg peak) with minimal exit dose, thereby reducing integral dose to surrounding tissues [424,425]. Comparative studies in pediatric medulloblastoma have shown PBT to be associated with lower rates of hearing loss, endocrine dysfunction, and secondary malignancies compared to photon-based therapy [426,427,428]. The unique dose distribution of PBT may inherently mitigate the radiation BEs by limiting low-dose exposure to non-target tissues [429,430]. Additionally, heavy ion RT, particularly carbon ion therapy, offers additional advantages [431]. With a higher relative biological effectiveness (RBE) and steeper dose gradients, carbon ions achieve superior tumor control while reducing normal tissue exposure [432,433]. Preclinical data suggest that carbon ion irradiation induces more robust immunogenic cell death and releases distinct DAMPs and cytokine profiles compared to photons, potentially altering radiation BEs in both tumor and normal tissues [434,435]. Early clinical outcomes in radioresistant tumors such as chordomas and sarcomas support these theoretical benefits, though rigorous studies assessing BEs modulation are still needed [436,437].

### 5.16. Spatially Fractionated and Lattice Radiotherapy

Spatially fractionated RT techniques, including lattice RT, aim to selectively irradiate sub-volumes within the tumor, leaving intervening regions unexposed [438,439]. This deliberate geometric heterogeneity reduces the irradiated volume and may consequently diminish the total pool of damaged cells capable of releasing BEs mediators [438]. Preclinical work has shown that lattice RT lowers systemic inflammatory responses compared to uniform whole-tumor irradiation, despite comparable tumor control [440,441]. Shielded tissues exhibit reduced γH2AX foci and oxidative damage, implicating decreased production of soluble bystander factors [183]. Mechanistically, sparing tumor sub-regions can mitigate mitochondrial ROS production and cfCh release into circulation, two well-characterized triggers of bystander signaling [183,270]. While lattice RT does not completely abolish systemic mediator release, it represents a promising procedural strategy for reducing BEs in partial-volume irradiation contexts.

### 5.17. Fractionation Schedules and Dose Rate Modulation

The dynamics of dose delivery profoundly influence BEs [44]. Single high-dose exposures are known to induce robust bystander responses, characterized by increased ROS, pro-inflammatory cytokines, and exosome-mediated signaling [61,68]. In contrast, hyper fractionation-dividing the total dose into multiple smaller fractions, can attenuate these effects [5]. Animal studies have demonstrated that fractionated irradiation reduces DNA damage biomarkers (e.g., γH2AX, 8-oxo-dG) and inflammatory mediators in shielded tissues compared to equivalent single doses [411]. In vitro, conditioned medium from cells exposed to fractionated regimens elicits significantly less genomic instability in naïve bystander cells, implicating both reduced mediator production and greater opportunity for clearance between fractions [442]. Mechanistically, fractionation allows irradiated cells time for DNA repair and resolution of stress responses, limiting the sustained activation of NFκB and TGF-β signaling that underpins radiation BEs propagation [314]. Similarly, dose rate modulation impacts bystander signaling [7]. Lower dose rates extend irradiation duration, potentially allowing ongoing mediator clearance, whereas high dose rates concentrate damage and amplify signal release [13]. This duality underscores the need for careful dose rate optimization to balance tumor control with suppression of deleterious NTEs.

### 5.18. Shielding Non-Targeted Tissues

While shielding to limit direct radiation exposure has long been routine in clinical RT, its specific role in mitigating radiation BEs has gained increasing traction [443]. Studies across multiple laboratories worldwide, including our own, show that strategic shielding of non-targeted regions during partial-body or focal irradiation significantly reduces systemic markers of oxidative stress, DNA damage, and inflammation in shielded tissues [103,444,445]. Although shielding does not block circulating mediators such as cfCh or cytokines, it prevents low-dose scatter that could otherwise act in concert with bystander signaling to promote genomic instability [117,311]. Emerging technologies in adaptive RT and motion gating further enhance this approach by minimizing incidental exposure to healthy tissues, thereby reducing stress-induced signal propagation from marginally irradiated zones [446,447,448].

### 5.19. Clinical Imperative and Technological Innovations

Integration of such germline variants into radio-genomic predictive models holds promise for tailoring RT protocols to individual patients [360,449]. High-risk genotypes could inform the selection of modalities with reduced integral dose, such as proton therapy or FLASH, or necessitate prophylactic administration of pharmacologic agents targeting bystander pathways [415,416,450]. Parallel efforts in radiomics and artificial intelligence (AI) have identified imaging-derived features predictive of BEs-related toxicities [451,452]. For example, textural heterogeneity in pre-treatment MRI and CT scans has been linked to subsequent development of fibrosis and neurocognitive decline, likely reflecting underlying microenvironmental susceptibilities to bystander signaling [453,454].

**Combination countermeasures:** As evidence for the clinical relevance of radiation BEs continues to mount, so too does the need for multi-pronged approaches that go beyond single-pathway interventions. Rather than targeting one mediator or delivery variable in isolation, emerging strategies seek to intercept bystander signaling at multiple nodes: production, propagation, and reception. These efforts reflect the complex, overlapping mechanisms that underpin BEs and acknowledge that no single modality is likely to offer complete protection across patients, tissues, or tumor types.

### 5.20. Converging on Exosome Biogenesis and Uptake

Exosomes serve as central vehicles of long-range bystander signaling, carrying DNA fragments, inflammatory miRNAs, and proteins from irradiated cells to distant recipients [337]. Blocking their biogenesis or uptake has shown promise in reducing untargeted damage [107]. Pharmacologic inhibition of neutral sphingomyelinase 2 (nSMase2), which mediates ceramide-dependent exosome release, reduces transmission of DNA damage and stress signals in both in vitro and animal models [455,456]. Downstream, receptor-mediated endocytosis of exosomes can be disrupted using agents like heparin or dynasore, which prevent internalization by target cells [457,458]. These inhibitors do not affect tumor response to direct radiation but do suppress γH2AX foci and senescence in unirradiated tissues exposed to irradiated-cell secretome [411,432]. Combining these inhibitors with standard therapies could selectively insulate normal tissues without compromising anti-tumor efficacy.

### 5.21. Degradation of cfCh and DAMPs

Beyond exosomes, a growing body of evidence implicates cfCh and DAMPs like HMGB1 as initiators of distant DNA damage [75,370]. These molecules, released by irradiated or dying cells, can integrate into the nuclei of non-targeted cells, activating cytosolic DNA sensors (e.g., cGAS/STING) and triggering inflammatory cascades [219,220]. Targeted enzymatic clearance, via recombinant DNase I or anti-HMGB1 antibodies, represents a promising route to blunt this arm of radiation BEs signaling [183,371]. In vivo, intravenous DNase I administered after partial-body irradiation reduces distant tissue γH2AX staining, micronuclei, and oxidative damage without impairing tumor response [372,373]. These findings suggest that intercepting extracellular DNA before it reaches susceptible cells can disrupt one of the most upstream elements of bystander propagation.

### 5.22. Dual Inhibition—Crosstalk Between Pathways

Recent studies underscore the benefit of simultaneously targeting more than one signaling vector [459]. For instance, combining mitochondrial-targeted antioxidants with gap junction blockades reduces both primary ROS production and intercellular propagation, yielding a more pronounced suppression of bystander markers [270,312]. Similarly, co-inhibition of exosome release and cfCh uptake attenuates senescence and inflammatory gene expression in distant tissues more effectively than either approach alone [117]. Such dual-inhibition paradigms mirror successful strategies in cancer therapy where mono-target agents fall short due to compensatory crosstalk [7]. In the context of BEs, the same redundancy is likely: blocking a single mediator may delay but not prevent RIGI unless multiple arms of the cascade are addressed in parallel.

### 5.23. Population-Level and Occupational Exposure Considerations

At the population level, awareness of radiation BEs has implications for minimizing unnecessary radiation exposure in diagnostic imaging and occupational settings [386,387]. Studies among interventional radiologists, nuclear plant workers, and astronauts exposed to low-dose cosmic radiation reveal systemic inflammatory markers and genomic instability consistent with radiation BEs, reinforcing the ALARA (As Low As Reasonably Achievable) principle across medical and industrial contexts [459,460,461,462,463,464,465,466,467].

Together, radiation BEs is not merely a side effect. It is a modifiable axis of RT biology, and in recognizing it as such, we open the door to both harm reduction and therapeutic gain.

### 5.24. Radiation BEs Is Not Always Bad

Radiation BEs has long been viewed primarily as a conduit of collateral damage, propagating oxidative stress, genomic instability, and carcinogenic risk beyond irradiated fields [58,468]. As research matured, a more nuanced narrative emerged: in certain contexts, bystander signaling appears capable of priming resilience, promoting repair, and mobilizing systemic defenses [59]. These paradoxical findings reflect the “unexpected turns” in radiation BEs, while their variability and fragility highlight the “falls” that have limited translation into clinical radiobiology [285]. Adaptive responses in shielded tissues were among the earliest surprises [86]. Partial-body irradiation studies revealed that shielded tissues sometimes upregulate antioxidant defenses [70,196]. For instance, rats with localized cranial exposure showed a 30–40% increase in SOD2 and catalase in the spleen, with reduced lipid peroxidation compared to whole-body irradiated or sham controls [196]. Similar protective signatures were reported in skin and lung models, where shielded regions expressed more Rad51 and methylation-regulating enzymes, suggesting systemic preconditioning [70]. However, these effects proved inconsistent [190]. Other studies under comparable conditions failed to reproduce antioxidant priming or reported exacerbated oxidative damage in bystander tissues [196,469]. This underscores how tightly adaptive BEs depends on dose, timing, and tissue context [190].

Immune modulation adds another layer of complexity [470]. While some studies suggest irradiated cells can sensitize tumor cells to immune-mediated killing, others show that conditioned media from irradiated cells may suppress NK cell cytotoxicity or have no clear effect [471,472]. These divergent outcomes highlight the unpredictable nature of bystander signaling in immune contexts. In vivo, localized irradiation increased systemic IL-12 and IFN-γ levels and CD8+ T-cell infiltration in shielded tissues [77,473]. These findings hint that bystander signaling may contribute to abscopal responses in RT [474]. Yet outcomes remain highly context dependent [475]. While immune activation can clear damaged or premalignant cells, it also risks fostering chronic inflammation and fibrosis [191,476,477,478,479,480]. The bystander-mediated immune effects are therefore a double-edged sword, shaped by microenvironmental and systemic factors [392].

Adding to this complexity are findings of epigenetic stability and/or destabilization in bystander cells [389]. Shielded tissues sometimes exhibit higher methyl-binding protein and histone silencing marker levels, repressing pro-inflammatory genes across cell generations, a transmissible “memory” of stress [208]. Sex-specific differences suggest that hormonal modulation adds variability [214]. But protective epigenetic signatures are not universal [390]. Other studies report global hypomethylation and chromatin disruption in bystander tissues, indicating genomic instability [44]. This duality reinforces the fragile balance of adaptive radiation BEs [68].

Translational and evolutionary implications stem from recognizing protective radiation BEs [44]. Controlled bystander signaling might precondition normal tissues to reduce off-target toxicities [125]. Its variability could help explain why some patients show unexpectedly low normal tissue damage in RT [391]. However, these concepts remain speculative. While pharmacologic suppression of harmful BEs has been explored, balancing the preservation of adaptive signaling with minimizing harm remains an unresolved challenge [315]. A more nuanced approach is needed, one that preserves adaptive signaling while minimizing harm [392]. Viewed evolutionarily, the dual nature of BEs may reflect a conserved stress response: injured regions send systemic signals to limit damage and prime resilience in distal tissues, echoing hormesis theories and suggesting potential adaptive value under certain conditions [393].

## 6. Experts’ Opinion

### 6.1. Radiation BEs Are Defined and Not Stochastic

Established as a genuine and significant phenomenon within radiobiology, substantial experimental data have demonstrated that cells not directly hit by IR can nevertheless exhibit radiation-like responses as a result of signals received from neighboring irradiated cells [52]. Evidence from in vitro cell cultures and in vivo models supports the reality of these effects, which have been consistently reproduced across various biological systems [58]. Cumulative research dispels earlier skepticism and underlines that BEs is a reproducible biological effect/phenomenon [44].

### 6.2. Radiation BEs Inflicts Systemic and Lasting Consequences

Radiation BEs extend biological effects beyond irradiated regions, reshaping the TME, driving off-target toxicities, and influencing long-term outcomes [44]. Current clinical models and treatment planning underestimate these systemic effects, particularly in spared tissues and vulnerable settings like pediatrics or reirradiation [121]. Recognizing BEs as a clinically relevant process is essential for improving risk assessment, biomarker development, and therapeutic decision-making in RT [89].

### 6.3. Radiation BEs Are Radiation-, System- and Disease-Specific

BEs are shaped by physical and disease-specific factors, including radiation quality and dose, dose rate, tissue type, and disease state [61]. Linear energy transfer, fractionation schedules, and tissue- or tumor-specific microenvironments alter bystander signaling, limiting uniform treatment strategies [44]. Addressing these variables is essential for translating BEs research into clinically effective approaches that minimize toxicity or enhance radiotherapy efficacy [44].

### 6.4. Molecular-Targeted Radioprotectors Are Superior over General/Global Agents for Radiation BEs

Radioprotective strategies for BEs should shift from broad-spectrum agents to molecularly targeted approaches that modulate key mediators like cytokines, ROS, and microRNAs [307]. Such interventions must selectively protect normal tissues without supporting tumor survival and require validation in preclinical models that accurately mimic the TME and clinical conditions [129]. Advancing these targeted therapies will rely on mechanistic insights and collaboration among molecular biologists, radiation oncologists, and pharmacologists to improve patient safety and treatment outcomes [106].

## 7. Conclusions

Radiation-induced BEs represent a transformative paradigm in radiobiology, challenging the long-held notion of RT as a strictly localized intervention. This review underscores the systemic and enduring consequences of ionizing radiation, revealing a complex web of intercellular signaling, immune modulation, epigenetic reprogramming, and metabolic rewiring that extends far beyond the irradiated field. The clinical implications are profound, manifesting as neurocognitive decline, cardiovascular disease, pulmonary fibrosis, gastrointestinal toxicity, and secondary malignancies, particularly in pediatric and long-term cancer survivors. Importantly, BEs are not merely a collateral phenomenon but a modifiable axis of RT biology. Emerging countermeasures, from COX-2 inhibitors and antioxidants to exosome blockers and FLASH RT offer promising avenues to mitigate off-target damage while preserving therapeutic efficacy. Moreover, the dual nature of BEs, capable of both harm and healing, invites a nuanced approach that balances suppression of detrimental effects with the harnessing of adaptive immune responses. Precision medicine, informed by radiogenomics and biomarker-driven surveillance, must integrate BEs into treatment planning and survivorship care. As RT technologies evolve, so too must our understanding of their systemic footprints. Recognizing BEs as defined, reproducible, and clinically relevant processes is imperative for advancing oncologic outcomes, minimizing toxicity, and improving quality of life. This review calls for a paradigm shift, one that embraces systemic awareness, molecular targeting, and personalized strategies to redefine the boundaries of modern radiation oncology.

## 8. Methodological Transparency: Criteria for the Selection of Studies

To ensure the integrity and relevance of this review, studies were identified and selected through a structured and transparent process. The objective was to synthesize current knowledge on both the biological and clinical aspects of radiation BEs and other NTEs. Literature searches were conducted in PubMed database using combinations of the following terms: radiation bystander effect, non-targeted radiation response, abscopal effect, RT toxicity, extracellular vesicles, gap junctions, oxidative stress, RPTs and immune modulation. Additional studies were identified through citation chaining from key reviews and pivotal primary research articles.

Studies were included if they met at least one of the following criteria: (i) provided mechanistic insights into BEs or NTEs using validated molecular or cellular assays, (ii) demonstrated reproducibility across experimental systems, including in vitro, in vivo, or patient-derived data, (iii) described clinically significant outcomes, particularly among pediatric or long-term cancer survivors, (iv) investigated countermeasures or mitigation strategies with translational potential, or (v) examined systemic effects of localized radiation therapy involving immune, epigenetic, or metabolic pathways. Studies were excluded if they lacked mechanistic depth or failed to differentiate BEs/NTEs from general radiation effects, relied solely on unvalidated or unreliable observations, omitted sufficient methodological details to assess reproducibility, or focused exclusively on radiation therapy modalities without addressing underlying biological effects.

In prioritizing the literature, prioritization was given to in vivo and clinical studies over purely in vitro investigations, as well as to experiments incorporating rigorous controls, dose–response analyses, and the integration of molecular signaling with functional or phenotypic endpoints. Although the review integrates a broad body of literature from early descriptive observations to modern mechanistic studies, ongoing advances continue to reshape our understanding of BEs and NTEs. Emerging mechanisms that remain speculative or insufficiently validated were included cautiously and clearly identified as hypothetical. This structured framework maintains focus on reproducible methodologies, biologically meaningful data, and mechanistic insights, thereby enhancing the relevance of the review for researchers and clinicians in radiation oncology.

## Figures and Tables

**Figure 1 cells-14-01761-f001:**
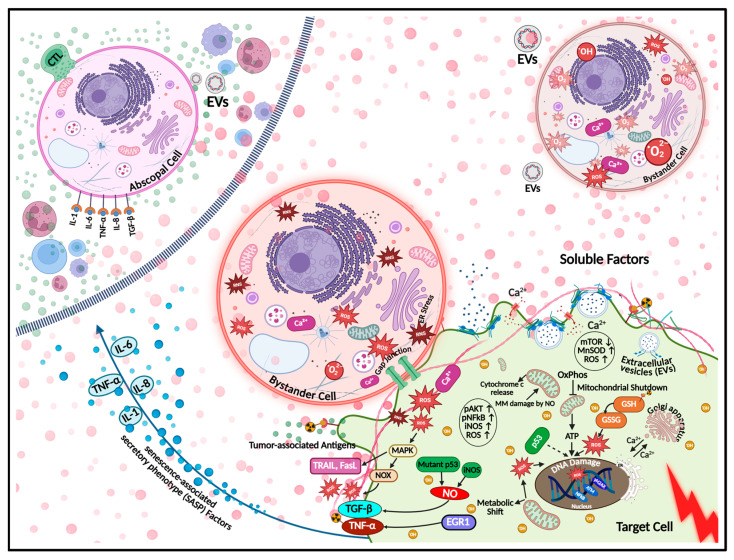
**Dynamics of Radiation BEs.** A schematic showing representative signaling flow-through and mechanisms how irradiated tumor cells trigger non-targeted effects by releasing soluble factors and extracellular vesicles, which affect neighboring, nearby, and distant (abscopal) cells, affecting their cellular and biological processes.

**Figure 2 cells-14-01761-f002:**
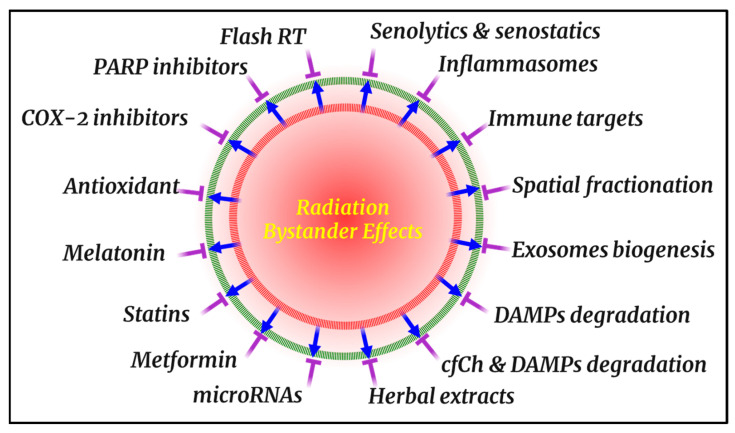
**Overview of potential countermeasures targeting radiation BEss.** Brief schema of evolving interventions targeting molecular determinants and cellular mechanisms underlying radiation BEs. These and other pipeline promising molecular targeted strategies are focused on mitigating off-target damage and enhancing the therapeutic index of radiation treatment, signifying the translational potential for improving patient outcomes in RT. Symbol explanation: Purple arrows denote inhibitory or counteractive interventions; blue arrows represent radiation bystander effects transmitted from irradiated to non-irradiated cells.

**Table 2 cells-14-01761-t002:** Genetic and molecular determinants, signaling pathways and cellular mechanisms involved in radiation.

Category	Mechanism of Action	Mediators	Ref.
Gap junction-mediated communication	Intercellular communication through gap junctions enables the transfer of signals between irradiated and bystander cells.	Connexins (Cx43), calcium ions, ROS, cytokines (TNF-α, IL-1α)	[88,122,128,135,138,164,195]
Soluble factor/cytokine signaling	Irradiated cells release cytokines and soluble factors that activate signaling pathways in bystander cells	Pro-inflammatory cytokines (TNF-α, IL-6, IL-8, IL-1, IL-33, TGF-β), Anti-inflammatory (IL-10), Growth factors (CSF2, VEGF), Signaling pathways (NFκB, STAT3, JAK-STAT, MAPK)	[85,100,110,128,134,135,136,137,138]
Oxidative stress and ROS/RNS signaling	Radiation-induced ROS and RNS propagate damage to bystander cells through direct diffusion or signaling cascades	ROS generators (NADPH oxidases NOX1-5, COX-2, mitochondrial dysfunction), RNS (NO, peroxynitrite via iNOS), Antioxidant suppression (SOD2, catalase, GST), Signaling (NFκB, MAPK, NLRP3 inflammasome)	[85,88,90,109,113,115,128,138,164,196,197,198,199]
NFκB-mediated signaling networks	NFκB activation serves as a central hub for multiple bystander signaling pathways, regulating survival, inflammation, and stress responses	NFκB components (RelA/p65, p50, IκBα), Downstream targets (Survivin, cIAP1/2, Bcl-2, MnSOD, hTERT), Feedback loops (TNF-α, MMP9, telomerase activation)	[89,91,92,95,100,102,104,114,122,138,198,199,200,201,202]
Extracellular vesicle/exosome-mediated transfer	Irradiated cells release EVs containing miRNAs, proteins, and other cargo that transfer signals to bystander cells	Exosomal miRNAs (miR-744-3p, miR-152, miR-21), Signaling (TLR4, DNA damage response), Cargo (ROS, nitric oxide, calcium signaling molecules)	[68,122,135,140,196,203]
MicroRNA regulation	Radiation alters miRNA expression in irradiated cells, with specific miRNAs propagating bystander effects	miR-21 (suppresses SOD2, promotes ROS and DNA damage), miR-663 (regulates TGF-β1 in feedback loops), let-7 family (stress response regulation), miR-30c, miR-144, miR-34a (epigenetic regulation)	[12,74,92,121,122,124,130,138,149,197,199,203,204]
TGF-β signaling pathways	TGF-β serves as a key mediator of BEs through both Smad-dependent and independent pathways	TGF-β/TGF-βRII signaling, Downstream (COX-2, iNOS, NADPH oxidases), Interactions (miR-21, miR-663, MAPK JNK/ERK), Tissue remodeling (Wnt/β-catenin, Notch signaling)	[95,109,120,124,125,128,130,134,148,149]
Mitochondrial dysfunction and signaling	Radiation disrupts mitochondrial function, leading to persistent ROS production and release of damage signals	Mitochondrial ROS, mtDNA release, Electron transport chain (ETC) dysfunction, Biophoton emission, exosome signaling, Crosstalk with nuclear DNA damage response	[89,90,91,92,109,113,114,115]
DNA damage response and repair signaling	DNA damage sensors and repair pathways in irradiated cells generate signals that propagate to bystander cells	DNA damage sensors (ATM, ATR, DNA-PK), Repair proteins (BRCA1, FANCD2, Chk1), Damage markers (γH2AX, 53BP1 foci), Checkpoint proteins (p53, p21)	[68,74,90,120,121,122,123,124,125]
Death receptor/apoptosis signaling	Irradiated cells release death ligands that induce apoptosis in bystander cells through receptor-mediated pathways	Death ligands (TRAIL, FasL, TNF-α), Receptors (TRAIL-R2 DR5, Fas, TNFR1), Apoptosis machinery (Caspases, PAR-4, Bcl-2 family), Survival signals (AKT, NFκB)	[114,137,138,151,163,205,206]
Autophagy modulation	Radiation affects autophagy pathways, influencing cell survival and bystander signaling	Autophagy markers (LC3-II, p62), Regulation (CSF2 signaling, rapamycin), Dual roles (Protective vs. pro-invasive effects)	[110,200]
Epigenetic modifications	Radiation induces epigenetic changes that can propagate BEs and genomic instability	DNA methylation (DNMTs, hypomethylation), Histone modifications (HDACs), miRNA regulation (miR-21, miR-30c, miR-144, miR-34a), Chromatin remodeling	[92,121,128,129,130]
Immune system modulation	Radiation alters immune cell function and signaling, affecting bystander responses through immune-mediated pathways	Immune cells (TAMs, T-cells, microglia), Immune signaling (TLR2/4/9, NLRs, DAMPs), Cytokines (IL-2, IL-10, TGF-β, TNF-α, IFN-γ), Adaptive responses (M1/M2 polarization)	[89,125,134,138,139,140,141]
Metabolic reprogramming	Radiation induces metabolic changes that contribute to BEs through altered energy metabolism and metabolite signaling	Glycolysis (GAPDH, LDHA, lactate production), Energy sensing (AMPK activation), Metabolite transport (MCT1), Microenvironment acidification	[95]
Stem cell and tissue microenvironment effects	Radiation affects stem cell niches and tissue microenvironments, influencing bystander cell behavior	Stem cell expansion (TGF-β, Notch pathway), Microenvironment remodeling (Stromal cells, ECM), Neural stem cells (Neurogenesis modulation), Hematopoietic effects (HSC dysfunction)	[128,134,141,144,151,163,207]
Systemic/abscopal effects	Localized radiation induces systemic signals that affect distant, non-irradiated tissues	Circulating factors (Cytokines, clastogenic factors), Systemic stress responses (Antioxidant pathways, Nrf2), Organ-specific effects (Heart, brain, bone marrow), Adaptive responses (Radioprotection, preconditioning)	[91,103,125,136,139,157]
Dose–response and temporal dynamics	BEs show specific dose–response relationships and temporal patterns	Low-dose specificity (Adaptive responses), Temporal waves (Biphasic responses, delayed effects), Feedback mechanisms (Adaptive inhibition), Spatial constraints (Distance-dependent effects)	[90,104,125,138,139,157,164]
Tissue-specific and cell-type-specific effects	Different cell types and tissues exhibit distinct bystander response patterns and mechanisms	Neural tissues (Glioblastoma, NSCs, astrocytes), Lung (Fibroblasts, epithelial cells), Breast (Mammary epithelial cells), Hematopoietic (HSCs, lymphoblasts)	[74,120,122,123,141,144,148,150,151,152,164]
Senescence-associated secretory phenotype (SASP)	Radiation-induced senescent cells secrete pro-inflammatory and growth-promoting factors, inducing bystander senescence through metabolic alterations.	SASP factors, AMPK/NFκB pathway activation, Glycolysis inhibition, Senolytic resistance pathways, Metabolic alterations	[142,143]
Ferroptosis-mediated BEs	Iron-dependent lipid peroxidation-driven cell death pathway with CD8+ T cell regulation and lipid peroxide transfer to bystander cells	Iron metabolism, Lipid peroxidation, CD8+ T cells, CCL5-SCD4 axis, B2M-TFRC axis, Lipid peroxide transfer	[116]
Enhanced metabolic reprogramming and bioenergetic alterations	Specific metabolite signaling and metabolic pathway disruption propagating to bystander cells through bioenergetic alterations	Lactate, Succinate, α-ketoglutarate, Itaconate, Glycolysis modulation, AMPK activation, Metabolic pathway disruption	[95,205,207]
Chromatin remodeling and histone modifications	Epigenetic modifications and chromatin remodeling complexes affecting gene expression and propagating bystander signals	Chromatin remodeling complexes, Histone modifications, Epigenetic modifications, Gene expression alterations	[44,208]
Intercellular calcium signaling networks	Gap-junction-mediated calcium ion cascades and small molecule transfer propagating bystander signals	Gap junctions, Calcium ion cascades, Nitric oxide, small molecule transfer, Calcium signaling pathways	[66,209]
Radiation-induced genomic instability	NTEs causing genomic instability in cells not directly irradiated through DNA damage and repair pathway disruption	Genomic instability, DNA damage, DNA repair pathway disruption, NTEs	[126,127,209]
Purinergic signaling networks	ATP and adenosine release from radiation-damaged cells activate purinergic receptors in bystander cells	ATP, Adenosine, P2X/P2Y receptors, A2A/A2B receptors, Calcium signaling cascades, Inflammatory responses	[210,211]
Complement system activation	Radiation-induced complement cascade activation, with complement proteins serving as bystander signals	Complement proteins (C3a, C5a), Complement cascade, Complement receptors, Inflammatory response propagation	[212]
Radiation-induced trained immunity	Epigenetic reprogramming in immune cells creates “trained immunity” states affecting bystander immune responses	Macrophages, Monocytes, Trained immunity, Epigenetic reprogramming, Metabolic reprogramming, Chromatin modifications	[213]
Circadian rhythm disruption	Radiation-induced circadian clock mechanism disruption with temporal signaling disruptions propagating to bystander cells	Circadian clock mechanisms, Temporal signaling, Circadian factors, Cell cycle regulation, DNA repair timing	[160]
Extracellular matrix (ECM) remodeling	Changes in ECM composition and matricellular proteins propagate mechanical and biochemical signals through integrin-mediated pathways	Thrombospondin, Osteopontin, Tenascin-C, Integrin-mediated pathways, Matricellular proteins	[153]
Cellular reprogramming factor release	Radiation-induced activation of pluripotency factors and developmental pathway molecules affecting bystander cell differentiation	Pluripotency factors, Developmental pathway molecules, Partial cellular reprogramming, Cell differentiation	[214]
Autophagy-lysosome pathway crosstalk	Dysregulated autophagy-lysosome pathways with secretion of lysosomal contents and autophagy-related proteins	Lysosomal contents, Autophagy-related proteins, Autophagy-lysosome pathway dysregulation	[110,215,216]
Non-coding RNA-based communication	In addition to miRNAs and extracellular vesicles, other non-coding RNAs (ncRNAs) such as long non-coding RNAs (lncRNAs) and circular RNAs (circRNAs) can modulate BEs by regulating DNA damage response, apoptosis, and intercellular signaling.	lncRNA HOTAIR, circRNAs, exosomal lncRNAs.	[111,131,132]
Hormetic and adaptive responses post low-dose RT	Hyper-radiosensitivity and induced radioresistance (HRS/IRR) show that not only low but also moderate doses can induce distinct adaptive BEs, both protective and detrimental, depending on the timing and microenvironment.	DNA repair pathways (Rad51, Ku70/80) and cell cycle checkpoint proteins.	[84,217]
Sex-specific and age-dependent BEs	BEs can vary by sex and age due to differences in hormonal environment, immune modulation, and antioxidant defenses.	Estrogen/testosterone, age-related senescence pathways, and altered cytokine profiles.	[214,218]
Microbiome and BEs interactions	The microbiome can influence and be influenced by BEs Via immune modulation, metabolite production, and gut–brain axis signaling after radiation exposure.	Microbial metabolites (short-chain fatty acids), microbial-derived ROS/NO, and TLR signaling.	[158,159]
Proteostasis and unfolded protein response (UPR)	Radiation-induced ER stress can trigger UPR in both irradiated and bystander cells, promoting inflammation, apoptosis, and potentially chronic BEs phenotypes.	PERK, ATF6, IRE1, chaperone proteins (GRP78/BiP).	[161,162]
Receptor tyrosine kinase signaling	Ligand-based activation of mitogenic/inflammatory pathways in bystander cells	EGFR, PDGFR, VEGFR, FGFR	[64,85]
cGAS–STING signaling	Cytosolic DNA detection leads to IFN and immune gene expression	cGAS, STING, IRF3, TBK1	[219,220]
Mitophagy dysregulation	Impaired mitochondrial turnover enhances ROS spread	PINK1, PARKIN, LC3B	[93,94]
Inflammasome activation	mtDNA or ROS triggers inflammasome-driven cytokine release	NLRP3, ASC, Caspase-1, IL-1β, IL-18	[221,222]
Mitochondria-associated endoplasmic reticulum membrane (MAM) disruption	Altered ER-mitochondria junctions affect Ca^2+^, ROS, apoptosis	IP3R, VDAC1, GRP75, MFN2	[223,224]
HIF-mediated hypoxic responses	Radiation-induced hypoxia drives pro-survival and angiogenic BEs	HIF-1α, VEGF, REDD1	[201,225,226]
RNA editing and dsRNA signaling	A-to-I RNA editing alters immune and stress signaling in bystander cells	ADAR1, PKR, RIG-I, IFN genes	[111,133]
Lipid mediator signaling	Radiation induces lipid remodeling that influences inflammation or apoptosis signaling.	Ceramide, S1P, eicosanoids (PGE2, LTB4)	[85,119]
Ion channel remodeling	Radiation alters voltage-gated ion channel activity	Kv (Voltage-gated potassium channels), Nav, Cl^−^ channels, Ca^2+^ channels	[166,167]
Endothelial bystander signaling	Angiocrine signals from irradiated endothelium affect nearby cells	Angiopoietin-2, E-selectin, ICAM-1	[145,146,147]
Protease/ECM enzyme activation	ECM degradation propagates pro-inflammatory and proinvasive signals	MMP2, MMP9, TIMP, matrikines	[154,155]
Heat shock protein signaling	Extracellular HSPs activate DAMP and immune sensor pathways	HSP70, HSP90, TLR4	[227,228]
YAP/TAZ mechanotransduction	Mechanical stress alters transcription and cellular adaptation	YAP, TAZ, RhoA, actin, integrins	[156]
Nitrosative stress	RNS signaling damage in bystander cells	ONOO^−^, iNOS, nitrotyrosine	[206,229]
Adenylate kinase signaling	Extracellular ATP/AMP affects purinergic and metabolic sensors	Adenylate kinase, CD73, AMPK	[59,95,96]
mTOR pathway signaling	Translational and nutrient signaling in bystander stress responses	mTORC1, S6K, eIF4E, TSC2	[97,118]
Ubiquitin-proteasome dysfunction	Altered protein degradation affects bystander signaling	RNF ligases, proteasomal subunits	[163,165]
Phagocytosis of irradiated debris	Engulfment of damaged cells triggers inflammatory cascades	DAMPs, phosphatidylserine, MerTK	[230,231,232]
Lipid rafts and membrane reorganization	Disruption affects receptor clustering, vesicle formation, and signal dissemination.	Caveolin-1, GM1, flotillin	[168,169,170,233]

**Table 3 cells-14-01761-t003:** Countermeasures for Radiation Bystander Effects: Evidence and Safety.

Countermeasure	* Evidence Level	Efficacy (Quantitative)	Safety Profile/Cautionary Note
Predictive modeling and SNP-guided planning (Precision/Personalized Countermeasures)	Emerging	Predictive modeling of BEs susceptibility	Requires validation; ethical considerations
mitochondrial antioxidants with gap junction blockers (Dual Inhibition Strategies)	Emerging	Synergistic suppression of BEs	Complexity in dosing, toxicity
NS-398, celecoxib (COX-2 Inhibitors)	High	~40% reduction in micronuclei formation	GI and cardiovascular toxicity with prolonged use
Avasopasem Manganese (GC4419) [superoxide dismutase (SOD) mimetic]	High	↓ oral mucositis severity	Favorable safety; FDA fast-tracked
Proton Therapy (Modality selection for reduced integral dose)	High	↓ secondary malignancies, normal tissue toxicity	Cost, access limitations
Shielding/Motion Gating (Physical exclusion of non-targeted tissues)	High	↓ oxidative stress, DNA damage	Standard practice; enhanced by adaptive RT
DNase I, CNPs, R-Cu (cfCh Degradation)	Low	↓ γ-H2AX, IL-6, chromosomal aberrations	DNase I: off-target risk; CNPs: immunogenicity; R-Cu: untested in humans
GW4869, miR-21 antagomirs (Exosome/miRNA Targeting)	Low to Moderate	↓ DNA damage, apoptosis in bystanders	GW4869: experimental; antagomirs: delivery challenges
Spatial Fractionation/Lattice RT	Low to Moderate	↓ systemic inflammation, γ-H2AX	Not widely implemented
Exosome Biogenesis/Uptake Inhibitors	Low to Moderate	↓ γ-H2AX, senescence in bystanders	Experimental; specificity concerns
DNase I, anti-HMGB1 (cfCh/DAMP Clearance)	Low to Moderate	↓ γ-H2AX, inflammation	DNase I: systemic effects; antibodies: immunogenicity
Melatonin (Pre-treatment Radioprotectants])	Moderate	↓ IL-6 and TNF-α in shielded organs	Favorable safety; crosses blood–brain barrier
Statins, Metformin, Herbal Extracts (Traditional radioprotectors)	Moderate	↓ ROS, DNA damage, cytokine release	Statins: myopathy risk; Metformin: GI upset; Herbal: variable
MitoQ, metformin (Mitochondrial/Metabolic Modulators)	Moderate	↓ γ-H2AX, apoptosis, oxidative stress	Mitochondrial inhibitors: cytotoxicity; timing critical
MCC950, Anakinra (Inflammasome/Immune Modulators)	Moderate	↓ IL-1β, pyroptosis, fibrosis	Risk of immunosuppression; timing essential
D+Q, navitoclax, rapamycin (Senolytics/Senostatics)	Moderate	↓ fibrosis, senescence markers	Navitoclax: thrombocytopenia; rapamycin: metabolic effects
Anakinra (IL-1 Receptor Antagonists)	Moderate	↓ pneumonitis, fibrosis	Potential immunosuppression
PARP, ATR inhibitors (DDR Inhibitors)	Moderate	↑ tumor response; possible BEs amplification	Tumor-specific targeting needed
FLASH RT	Moderate	↓ fibrosis, neuroinflammation, BEs markers	Delivery challenges; under investigation
Carbon Ions (Heavy Ion Therapy)	Moderate	↑ tumor control; ↓ normal tissue dose	LET-specific toxicity; limited availability
Hyperfractionation and dose Rate Modulation (Temporal control of radiation delivery)	Moderate	↓ BEs markers with hyperfractionation	Requires precise planning
NAC, Vitamin E (Antioxidants)	Moderate to High	Reduced γ-H2AX foci and lipid peroxidation	Generally safe; tumor protection risk debated
Alpha-emitter (^225^Ac-NM600, ^223^Ra dichloride), Beta-emitter (^177^Lu-PSMA-617, ^131^I), combination therapy (RPT + anti-PD-1 antibodies) (Radiopharmaceuticals therapy)	Moderate	α-emitter RPT ↓ Tregs and ↑ activated CD8+ T cells; clinical trials report immune activation and survival benefits	Complex systemic biodistribution; antioxidants may interfere with immune priming; LET-tailored testing required
2-Deoxyglucose (Glycolysis Inhibitor)	Low	Disrupts metabolic coupling	Cytotoxicity concerns
Apyrase, Suramin (Purinergic Signaling Inhibitors)	Experimental	↓ ATP-mediated BEs signaling	Limited data
Disulfiram (Gasdermin D inhibitor)	Emerging	Blocks pyroptosis and cytokine release	Known drug; repurposing potential
Amifostine (Thiol-based free radical scavenger)	Moderate	↓ oxidative stress; radioprotection in preclinical models	Limited clinical uptake due to toxicity and inconsistent efficacy

↑ indicates increase; ↓ indicates decrease. * Selectively based on documented experimentally observed studies. It is not a comparison between the countermeasure deliverables.

## Data Availability

All data and materials are available in the main text of the manuscript.

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
