# Peer review of "Radiation Without Borders: Unraveling Bystander and Non-Targeted Effects in Oncology"

_cells, 2025, doi:10.3390/cells14221761_

Round 1
Reviewer 1 Report (Previous Reviewer 1)
Comments and Suggestions for Authors
The authors were responsive to my critique. No more questions
Author Response
Comments by reviewer 1:
The authors were responsive to my critique. No more questions
We appreciate the reviewer’s positive feedback.
Reviewer 2 Report (New Reviewer)
Comments and Suggestions for Authors
The manuscript aims to synthesize current molecular, cellular and clinical evidence on radiation-induced bystander effects (BE) and other non-targeted effects (NTEs), explain mechanisms (gap junctions, soluble mediators, EVs, oxidative stress, mitochondria, epigenetics, immune modulation), describe clinical consequences (neurocognitive decline, cardiovascular disease, fibrosis, secondary cancers), and to survey potential countermeasures and translational strategies to reduce harmful off-target effects while highlighting beneficial (adaptive/immune) BE.
This is a narrative review that aggregates: in-vitro mechanistic studies (microbeam and medium-transfer assays), in-vivo animal models (partial-body irradiation and shielding experiments), and clinical/epidemiologic evidence (survivorship cohorts and trials). The authors organize evidence by mechanism, radiation quality/delivery (LET, dose rate, FLASH), organ systems, translational countermeasures and future directions.
The authors conclude that BE/NTE are reproducible, clinically relevant phenomena that require shifting radiotherapy planning from a purely local view to a systemic one. They call for: (i) incorporation of BE into personalized planning and survivorship surveillance, (ii) development of molecularly targeted radioprotectors and combination countermeasures, (iii) prioritization of translational preclinical models and biomarker development, and (iv) cautious optimization of high-LET and internal delivery modalities with dedicated risk–benefit assessment.
This is a high-quality, timely, and largely comprehensive narrative review that would be very valuable to clinicians and translational scientists. It successfully synthesizes mechanisms and translational approaches and is up-to-date.
I have nevertheless some comments in order to improve this manuscript:
1) add methodological transparency about how studies were selected
2) expand on radiopharmaceutical (RPT)-specific BE and microdosimetry:
The review discusses abscopal effects and immune-mediated beneficial BEs mainly in the context of external beam RT combined with immunotherapy. But it does not provide a focused treatment of systemic radiopharmaceutical therapies and their unique ability to drive systemic immune responses or positive BEs. Their pharmacokinetics (circulation, tumor uptake, off-target retention) and microdistribution create complex exposure patterns (low-dose host tissues + very high dose in targeted tumor cells). These features uniquely shape BE and immune outcomes.
The manuscript should add a dedicated subsection titled e.g. “Radiopharmaceutical therapy: high-LET microdosimetry, systemic biodistribution, and BE/abscopal biology” covering: agent classes (β vs α), examples (177Lu, 225Ac, 223Ra, 131I), mechanistic pathways (ICD, DAMPs, exosomes, cfCh), preclinical/clinical reports of abscopal responses, and suggested monitoring/mitigation strategies. Currently that topic is not adequately covered in the text.
3) provide a clearer evidence-grading/quantitative element (tables or risk grading) and explicit cautionary notes on countermeasure safety. After those targeted additions, the manuscript will be stronger and more useful for direct clinical translation and for guiding future research in high-LET and radiopharmaceutical contexts.
4) Some countermeasures effective for low-LET (e.g., global antioxidants) may be less effective against clustered lesions or may interfere with desired immune priming. The review lightly touches on dual inhibition but should recommend LET-specific preclinical testing.
Author Response
Comments by reviewer 2:
1) add methodological transparency about how studies were selected
We have added a dedicated subsection detailing our study selection process. Specifically, we describe the databases searched, inclusion/exclusion criteria, and the rationale for selecting studies spanning in vitro mechanistic work, in vivo preclinical models, and clinical/epidemiologic reports. This now clarifies the structured approach we used to aggregate evidence while maintaining a narrative review format. The revised content is now included on page 35, section 8.
2) expand on radiopharmaceutical (RPT)-specific BE and microdosimetry: The review discusses abscopal effects and immune-mediated beneficial BEs mainly in the context of external beam RT combined with immunotherapy. But it does not provide a focused treatment of systemic radiopharmaceutical therapies and their unique ability to drive systemic immune responses or positive BEs. Their pharmacokinetics (circulation, tumor uptake, off-target retention) and microdistribution create complex exposure patterns (low-dose host tissues + very high dose in targeted tumor cells). These features uniquely shape BE and immune outcomes. The manuscript should add a dedicated subsection titled e.g. “Radiopharmaceutical therapy: high-LET microdosimetry, systemic biodistribution, and BE/abscopal biology” covering agent classes (β vs α), examples (177Lu, 225Ac, 223Ra, 131I), mechanistic pathways (ICD, DAMPs, exosomes, cfCh), preclinical/clinical reports of abscopal responses, and suggested monitoring/mitigation strategies. Currently that topic is not adequately covered in the text.
We have added a new subsection titled “Radiopharmaceutical therapy: high-LET microdosimetry, systemic biodistribution, and BE/abscopal biology”. This section covers:
- Agent classes (β-emitters vs α-emitters) and representative examples (¹⁷⁷Lu, ²²⁵Ac, ²²³Ra, ¹³¹I)
- Pharmacokinetics and microdistribution patterns that create heterogeneous exposures in host tissues versus targeted tumor cells
- Mechanistic pathways including immunogenic cell death (ICD), DAMPs, exosomes, and circulating cfDNA/Chromatin fragments
- Preclinical and clinical evidence for abscopal responses and systemic Bes
- Suggested monitoring strategies and potential mitigation approaches
The revised content is now included on page 27, section 5, subsection 5.12.
3) provide a clearer evidence-grading/quantitative element (tables or risk grading) and explicit cautionary notes on countermeasure safety. After those targeted additions, the manuscript will be stronger and more useful for direct clinical translation and for guiding future research in high-LET and radiopharmaceutical contexts.
We have incorporated summary tables highlighting the strength of evidence for BE/NTE across different radiation modalities and mechanistic pathways. Explicit cautionary notes regarding countermeasure safety have been added.
The revised content is now included on page 32, section 5.
4) Some countermeasures effective for low-LET (e.g., global antioxidants) may be less effective against clustered lesions or may interfere with desired immune priming. The review lightly touches on dual inhibition but should recommend LET-specific preclinical testing.
We have expanded discussion on countermeasures, emphasizing that agents effective in low-LET contexts (e.g., global antioxidants) may be less effective or could interfere with immune-mediated beneficial effects in high-LET or clustered lesion settings. We explicitly recommend LET-specific preclinical testing for candidate radioprotectors and combination strategies to ensure safety and efficacy.
The revised content is now included on page 28, section 5, subsection 5.12.
This manuscript is a resubmission of an earlier submission. The following is a list of the peer review reports and author responses from that submission.
Round 1
Reviewer 1 Report
Comments and Suggestions for Authors
Ramamurthy et al. is a well-written and highly comprehensive review on radiation bystander effects (BSE) and non-targeted effects (NTEs) in the context of radiotherapy. The authors systematically cover molecular mechanisms, clinical implications, and potential countermeasures. The article demonstrates significant breadth and depth, which makes it a valuable contribution to the field. However, some areas need improvement for clarity, conciseness, and critical evaluation.
- The manuscript is very dense; long sentences and heavy jargon reduce readability.
- Some sections (especially Table 2 with pathways and mediators) are overloaded with details, which could overwhelm general readers. Consider condensing into focused subsections with summary tables.
- While the review is thorough, parts are read like an encyclopedic list of pathways rather than a critical synthesis.
- Authors should provide stronger critical commentary on which mechanisms are best established vs. still speculative?
- The paper currently presents the field as if all studies are in alignment. However, radiobiology literature contains conflicting reports that should be acknowledged.
Some areas of the review article are dense, as suggested in the critique to the authors that need to be addressed.
Author Response
Manuscript ID: cells-3870543
Type: Review
Title: The Unseen Fallout: Bystander Responses of Cancer Radiotherapy in Human Health
Authors: Madhi Oli Ramamurthy, Poorvi Subramanian, Sivaroopan Aravindan, Loganayaki Periyasamy, Natarajan Aravindan *
Author’s Response to Reviews:
We truly thank the reviewers for their insightful review and valuable suggestions. We have addressed each reviewers’ concerns in the revised manuscript. Please see below for the point-by-point response to the reviewer’s critiques. All changes in the revised manuscript are indicated in red font.
Reviewer 1:
- Ramamurthy et al. is a well-written and highly comprehensive review on radiation bystander effects (BSE) and non-targeted effects (NTEs) in the context of radiotherapy. The authors systematically cover molecular mechanisms, clinical implications, and potential countermeasures. The article demonstrates significant breadth and depth, which makes it a valuable contribution to the field.
We thank the reviewer for appreciating the content and significance of this review.
- The manuscript is very dense; long sentences and heavy jargon reduce readability.
We appreciate the reviewer’s insights on the manuscript presentation. As advised, the manuscript has been carefully revised, addressing all these concerns. Since the changes are throughout the manuscript, the edits are not highlighted in the text.
- Some sections (especially Table 2 with pathways and mediators) are overloaded with details, which could overwhelm general readers. Consider condensing into focused subsections with summary tables.
We value the reviewer's recommendation. We would like to respectfully draw attention to the fact that Table 2 contains essential information, including the type of bystander response, the mode of action, the drivers involved, and the citations. We believe that this understanding is essential to this review and will give readers the information they need in a nutshell. Nonetheless, we concur with the critic that the general readership may occasionally be overwhelmed by the tabular compilation. With new structured text and distinct subsections (Sections 3.6 through 3.10) classifying the information as documented, in investigation, or under development, we have improved the navigability of these inferences. This allows readers to contextualize the information without sacrificing the depth of the mechanistic content.
This new information is now included in:
Sections – 3.6 through 3.10
Page 12 Paragraph 3 through Page 15 Paragraph 1
- While the review is thorough, parts are read like an encyclopedic list of pathways rather than a critical synthesis.
We have updated the manuscript as suggested, adding organized content and separate subsections (Sections 3.6 through 3.10) that provide a vital synthesis of the molecular signaling pathways, the manner of action, and the types of bystander responses. Additionally, the core of these conclusions was categorized as realized, under inquiry, or in development. Please refer to the answers to the previous criticism.
Sections – 3.6 through 3.10
Page 12 Paragraph 3 through Page 15 Paragraph 1
- Authors should provide stronger critical commentary on which mechanisms are best established vs. still speculative?
We agree with the reviewer. As discussed above (answers to the reviewer’s critiques 3 and 4), the revised manuscript now includes structured text with explicitly categorizing (Section 3.6) the information as documented, moderately supported, or under investigation. Focusing on restrictions, context-dependency, regions needing confirmation, and supporting citations, the new information also explains the reasoning behind their classification.
- The paper currently presents the field as if all studies are in alignment. However, radiobiology literature contains conflicting reports that should be acknowledged.
We completely agree with the reviewer that not all radiation bystander effects are linear and /or detrimental. To that end, we would like to humbly point out that such discrepancies are iterated throughout the revised manuscript, particularly recognizing limitations such as radiation quality, quantity, experimental systems, and/or time-associated dynamic signaling (complementing, compromising, and opposite) events. Further, the limitation in translating, in vitro systems realized bystander responses to the preclinical in vivo or in clinical settings due to the complex interplay of stromal elements, immune response, and heterogeneous tissue architecture are iterated in the revised manuscript. Also, we would like to bring to the attention that a separate section discussing the ‘radiation bystander effect is not always bad’ is available in the manuscript, indicating that not all studies or observations are aligned.
Page 3 Paragraph 4 through Page 4 Paragraph 1
Page 12 Paragraph 3 through Page 15 Paragraph 1
Page 31, Paragraph 1 Section 5.22
Reviewer 2 Report
Comments and Suggestions for Authors
This manuscript is well-written and has the potential to be a seminal review in the field. The authors are encouraged to revise this manuscript by incorporating the following points that will significantly improve its clarity and importance in the field:
- The acronym “BSE” is not a standard in the field. However, “Bystander Effects” is commonly abbreviated as “BE” or “RIBE” (Radiation-Induced Bystander Effects). For clarity and adherence to the field-specific terminology, I strongly recommend using “BE” or “RIBE” throughout the manuscript.
- The review correctly identifies the clinical relevance of BE. However, a significant portion of the foundational research on BE has been conducted in vitro, and there are often discrepancies between these findings and those from in vivo and clinical studies; these discrepancies arise due to the complexities associated with the in vivo environment (see for example refs. 87, 92 in the manuscript). Now, to provide a balanced and scientifically rigorous review, the authors should discuss these discrepancies, particularly in the introduction or a dedicated section on model systems.
- The authors should provide a clear rationale and discuss how the in vitro findings, despite their limitations, have contributed to our mechanistic understanding.
- Prioritizing the findings from animal models and human studies would be important, as these are more directly translatable to the clinical context.
- A very important point of this review is that it focuses on “ionizing radiation” but does not differentiate between various radiation qualities. Given the known differences in biological effects, a discussion on the influence of radiation quality (e.g., low-LET photons vs. high-LET protons and carbonions) on the manifestation of bystander effects is essential. High-LET radiation, in particular, is known to induce a more potent and distinct bystander response. Including this discussion would significantly enhance the quality (scientific depth), appeal, and relevance of this paper to modern RT modalities.
- This review also introduces FLASH RT, a high-dose-rate modality, as a countermeasure. This is a very critical point. A more detailed discussion on the influence of dose-rate on BE would be highly beneficial. The author should look at the publication “https://doi.org/10.3390/dna4010002”. The differences between conventional RT and FLASH RT are not limited to dose-rate but also include potential changes in oxygen depletion, radical formation, and the subsequent biological cascade, all of which could influence BE. A dedicated section on how dose-rate modulates bystander responses would strengthen this part of the review.
Author Response
Manuscript ID: cells-3870543
Type: Review
Title: The Unseen Fallout: Bystander Responses of Cancer Radiotherapy in Human Health
Authors: Madhi Oli Ramamurthy, Poorvi Subramanian, Sivaroopan Aravindan, Loganayaki Periyasamy, Natarajan Aravindan *
Author’s Response to Reviews:
We truly thank the reviewers for their insightful review and valuable suggestions. We have addressed each reviewers’ concerns in the revised manuscript. Please see below for the point-by-point response to the reviewer’s critiques. All changes in the revised manuscript are indicated in red font.
Reviewer 2:
- This manuscript is well-written and has the potential to be a seminal review in the field.
We thank the reviewer for appreciating the content and significance of this review.
- The acronym “BSE” is not a standard in the field. However, “Bystander Effects” is commonly abbreviated as “BE” or “RIBE” (Radiation-Induced Bystander Effects). For clarity and adherence to the field-specific terminology, I strongly recommend using “BE” or “RIBE” throughout the manuscript.
As advised, the abbreviation for bystander effect has been corrected to BE all throughout the manuscript.
- The review correctly identifies the clinical relevance of BE. However, a significant portion of the foundational research on BE has been conducted in vitro, and there are often discrepancies between these findings and those from in vivo and clinical studies; these discrepancies arise due to the complexities associated with the in vivo environment (see for example refs. 87, 92 in the manuscript). Now, to provide a balanced and scientifically rigorous review, the authors should discuss these discrepancies, particularly in the introduction or a dedicated section on model systems.
We completely agree with the reviewer, and the revised manuscript now includes the significance of in vitro studies (Section 3.7, Page 13, last paragraph to Page 14 first paragraph). Furthermore, the limitations in translating in vitro systems-realized bystander responses to preclinical in vivo (or clinical settings) due to the complex interplay of stromal elements, immune response, and heterogeneous tissue architecture are iterated in the revised manuscript. Also, we would like to bring to the attention that a separate section discussing the ‘radiation bystander effect is not always bad’ is available in the manuscript, indicating that not all studies or observations are aligned.
Introduction - Page 3 Paragraph 4 through Page 4 Paragraph 1
Page 12 Paragraph 3 through Page 15 Paragraph 1
Page 31, Paragraph 1 Section 5.22
- The authors should provide a clear rationale and discuss how the in vitro findings, despite their limitations, have contributed to our mechanistic understanding.
We appreciate the suggestion. As advised, the rationale and the impact of in vitro findings are now included in the revised manuscript (Section 3.7).
Page 13, last paragraph – Page 14, first paragraph.
- Prioritizing the findings from animal models and human studies would be important, as these are more directly translatable to the clinical context.
We concur with the reviewer entirely. The amended text now includes, as indicated, the importance of animal and/or human research (as opposed to the in vitro systems approach) to show the biological relevance of radiation bystander reaction and to convert the results to the clinic (section 3.8).
Page 14, paragraph 2.
- A very important point of this review is that it focuses on “ionizing radiation” but does not differentiate between various radiation qualities. Given the known differences in biological effects, a discussion on the influence of radiation quality (e.g., low-LET photons vs. high-LET protons and carbonions) on the manifestation of bystander effects is essential. High-LET radiation, in particular, is known to induce a more potent and distinct bystander response. Including this discussion would significantly enhance the quality (scientific depth), appeal, and relevance of this paper to modern RT modalities.
We apologize for the oversight and completely agree with the reviewer on the significance of iterating the BE differences with differences in radiation quality. As suggested, this information in detail is now included in the revised manuscript (section 3.9).
Page 14, last paragraph.
- This review also introduces FLASH RT, a high-dose-rate modality, as a countermeasure. This is a very critical point. A more detailed discussion on the influence of dose-rate on BE would be highly beneficial. The author should look at the publication “https://doi.org/10.3390/dna4010002”. The differences between conventional RT and FLASH RT are not limited to dose-rate but also include potential changes in oxygen depletion, radical formation, and the subsequent biological cascade, all of which could influence BE. A dedicated section on how dose-rate modulates bystander responses would strengthen this part of the review.
We appreciate the reviewer’s suggestion. The revised manuscript now includes the discussion on the influence of dose rate on BE (section 3.10).
Page 15, paragraph 1.
Reviewer 3 Report
Comments and Suggestions for Authors
The article "The Unseen Fallout: Bystander Responses of Cancer Radiotherapy in Human Health" is a review of molecular mechanisms widely known in the field of biochemistry (not necessarily associated with radiation) and mentions some well-known late effects of radiation. Therefore, the title and objective of the manuscript are inconsistent with its content.
For example, it is well-known that late effects of radiotherapy include health problems that arise months or years after treatment. These effects can include fibrosis, cardiovascular problems, skin changes, gastrointestinal problems, urinary difficulties, fertility and sexual disorders, and an increased risk of secondary cancers. The occurrence and severity of these effects vary depending on the individual, the type of cancer, and the area of the body treated.
In Table 1 of the article, the authors list deterministic effects and late effects of radiation that have been widely known since radiation began to be used in medical practice. These effects are associated with the absorbed radiation dose received, the area of the body treated, and the patient's own characteristics. The title "Table 1. Class of risks associated with radiation-bystander and/or abscopal effects” is completely erroneous.
Late effects of radiation therapy are well documented in the medical literature and are not "unseen fallout."
Conversely, it is well known that molecular processes are dynamic and biological processes are based on a complex network of intercellular communication involving soluble factors, such as cytokines, growth factors, chemokines, and hormones. The authors review several molecular mechanisms widely described in biochemical literature, which they erroneously title “Molecular Mechanisms of the Bystander Effect.”
Author Response
Manuscript ID: cells-3870543
Type: Review
Title: The Unseen Fallout: Bystander Responses of Cancer Radiotherapy in Human Health
Authors: Madhi Oli Ramamurthy, Poorvi Subramanian, Sivaroopan Aravindan, Loganayaki Periyasamy, Natarajan Aravindan *
Author’s Response to Reviews:
We truly thank the reviewers for their insightful review and valuable suggestions. We have addressed each reviewers’ concerns in the revised manuscript. Please see below for the point-by-point response to the reviewer’s critiques. All changes in the revised manuscript are indicated in red font.
Reviewer 3:
- The article "The Unseen Fallout: Bystander Responses of Cancer Radiotherapy in Human Health" is a review of molecular mechanisms widely known in the field of biochemistry (not necessarily associated with radiation) and mentions some well-known late effects of radiation. Therefore, the title and objective of the manuscript are inconsistent with its content. For example, it is well-known that late effects of radiotherapy include health problems that arise months or years after treatment. These effects can include fibrosis, cardiovascular problems, skin changes, gastrointestinal problems, urinary difficulties, fertility and sexual disorders, and an increased risk of secondary cancers. The occurrence and severity of these effects vary depending on the individual, the type of cancer, and the area of the body treated.
We apologize for the oversight, and as advised the manuscript is carefully revised to remove the signaling mechanisms and citations that are beyond the central scope. However, as the focus of this review is centered exclusively on radiation bystander effect (not the direct effects of radiation), uniquely synthesizing the mechanisms involved, compiling the divergence with experimental models, qualities of radiation etc., clinical implications realized, and the probable countermeasures, we believe the title matches the objectives of the review.
- In Table 1 of the article, the authors list deterministic effects and late effects of radiation that have been widely known since radiation began to be used in medical practice. These effects are associated with the absorbed radiation dose received, the area of the body treated, and the patient's own characteristics. The title "Table 1. Class of risks associated with radiation-bystander and/or abscopal effects” is completely erroneous.
The title for table 1 is corrected and now reads as “Table 1. Class of risks associated with off-target effects of radiation.”
- Late effects of radiation therapy are well documented in the medical literature and are not "unseen fallout." Conversely, it is well known that molecular processes are dynamic and biological processes are based on a complex network of intercellular communication involving soluble factors, such as cytokines, growth factors, chemokines, and hormones. The authors review several molecular mechanisms widely described in biochemical literature, which they erroneously title “Molecular Mechanisms of the Bystander Effect.”
Please see the response to the critique one.
Round 2
Reviewer 2 Report
Comments and Suggestions for Authors
The authors convincingly addressed all my critical remarks.
- The acronym “BSE” is not a standard in the field. However, “Bystander Effects” is commonly abbreviated as “BE” or “RIBE” (Radiation-Induced Bystander Effects). For clarity and adherence to the field-specific terminology, I strongly recommend using “BE” or “RIBE” throughout the manuscript. The Authors addressed this comment.
- The review correctly identifies the clinical relevance of BE. However, a significant portion of the foundational research on BE has been conducted in vitro, and there are often discrepancies between these findings and those from in vivo and clinical studies; these discrepancies arise due to the complexities associated with the in vivo environment (see for example refs. 87, 92 in the manuscript). Now, to provide a balanced and scientifically rigorous review, the authors should discuss these discrepancies, particularly in the introduction or a dedicated section on model systems. The Authors addressed this comment.
- The authors should provide a clear rationale and discuss how the in vitro findings, despite their limitations, have contributed to our mechanistic understanding. The Authors addressed this comment.
- Prioritizing the findings from animal models and human studies would be important, as these are more directly translatable to the clinical context. The Authors addressed this comment.
- A very important point of this review is that it focuses on “ionizing radiation” but does not differentiate between various radiation qualities. Given the known differences in biological effects, a discussion on the influence of radiation quality (e.g., low-LET photons vs. high-LET protons and carbonions) on the manifestation of bystander effects is essential. High-LET radiation, in particular, is known to induce a more potent and distinct bystander response. Including this discussion would significantly enhance the quality (scientific depth), appeal, and relevance of this paper to modern RT modalities. The Authors addressed this comment.
- This review also introduces FLASH RT, a high-dose-rate modality, as a countermeasure. This is a very critical point. A more detailed discussion on the influence of dose-rate on BE would be highly beneficial. The author should look at the publication “https://doi.org/10.3390/dna4010002”. The differences between conventional RT and FLASH RT are not limited to dose-rate but also include potential changes in oxygen depletion, radical formation, and the subsequent biological cascade, all of which could influence BE. A dedicated section on how dose-rate modulates bystander responses would strengthen this part of the review. The Authors addressed this comment.
Author Response
Thank you
Reviewer 3 Report
Comments and Suggestions for Authors
Indeed, the molecular mechanisms observed in vitro often differ from those observed in vivo or in clinical settings. This is due to the complex interplay of stromal elements, immune responses, and heterogeneous tissue architectures present in living organisms. However, the molecular mechanisms described by the authors in the article entitled "The Unseen Fallout: Bystander Responses of Cancer Radiotherapy in Human Health” are not unique to radiation. In my opinion, the article does not provide a review or discussion of relevant concepts or specific future directions related to the bystander effect. This review examines widely recognized molecular mechanisms in the field of biochemistry (not necessarily related to radiation) and highlights some well-known late effects of radiation. Therefore, the title and objective of the manuscript are inconsistent with its content.
Author Response
Thank you